# LiSBOA: LiDAR Statistical Barnes Objective Analysis for optimal design of LiDAR scans and retrieval of wind statistics. Part I: Theoretical framework

Stefano Letizia, Lu Zhan, and Giacomo Valerio Iungo

Wind Fluids and Experiments (WindFluX) Laboratory, Mechanical Engineering Department, The University of Texas at Dallas, 800 W Campbell Rd, 75080 Richardson, TX, USA

**Correspondence:** Giacomo Valerio Iungo, email: valerio.iungo@utdallas.edu

**Abstract.** A LiDAR Statistical Barnes Objective Analysis (LiSBOA) for the optimal design of LiDAR scans and retrieval of the velocity statistical moments is proposed. The LiSBOA represents an adaptation of the classical Barnes scheme for the statistical analysis of unstructured experimental data in $N$-dimensional space and it is a suitable technique for the evaluation over a structured Cartesian grid of the statistics of scalar fields sampled through scanning LiDARs. The LiSBOA is validated and characterized via a Monte Carlo approach applied to a synthetic velocity field. This revisited theoretical framework for the Barnes objective analysis enables the formulation of guidelines for the optimal design of LiDAR experiments and efficient application of the LiSBOA for the post-processing of LiDAR measurements. The optimal design of LiDAR scans is formulated as a two cost-function optimization problem including the minimization of the percentage of the measurement volume not sampled with adequate spatial resolution and the minimization of the error on the mean of the velocity field. The optimal design of the LiDAR scans also guides the selection of the smoothing parameter and the total number of iterations to use for the Barnes scheme. The LiSBOA is assessed against a numerical dataset generated using the virtual LiDAR technique applied to the data obtained from a large eddy simulation (LES). The optimal sampling parameters for a scanning Doppler pulsed wind LiDAR are retrieved through the LiSBOA, then the estimated statistics are compared with those of the original LES dataset, showing a maximum error of about 4% for both mean velocity and turbulence intensity.

## 15 List of symbols

- $x, y, z$: streamwise, spanwise, vertical Cartesian coordinates

- $t$: non-spatial coordinate or time

- $u, v, w$: streamwise, spanwise, vertical velocity components

- $u_{\mathrm{LOS}}$: radial or line-of-sight wind speed

- $L$: number of realizations/scans

- $\theta$: azimuth angle

- $\beta$: elevation angle

- $\Delta\theta$: azimuth-angle resolution

- $\Delta\beta$: elevation-angle resolution

- $\tau_a$: accumulation time

- $\Delta r$: gate length

- $N_r$: number of range gates along the laser beam

- $T$: total sampling time

- $\sigma$: smoothing parameter

- $m$: number of iterations

- $R_{\mathrm{max}}$: radius of influence

- $\boldsymbol{\Delta n}$: half-wavelength vector

- $\boldsymbol{\Delta n_0}$: fundamental half-wavelength vector

- $\Delta d$: random data spacing

- $\boldsymbol{dx}$: resolution vector in Cartesian coordinates

- $D^m$: response at the $m$-th iteration

- $\boldsymbol{r_i}$: position in the N-dimensional space of the i-th grid node of the Cartesian grid

- $\boldsymbol{r_j}$: position in the N-dimensional space of the j-th sample

- $\epsilon^I$: cost function I (data loss)

- $\epsilon^{II}$: cost function II (standard deviation of the sample mean)

- $N_i$: total number of nodes of the Cartesian grid

- $\tau$: integral time-scale

- $\tilde{\cdot}$: spatial variable in the scaled frame of reference

# 1 Introduction

Reliable measurements of the wind-velocity vector field are essential to understand the complex nature of atmospheric turbulence and provide valuable datasets for the validation of theoretical and numerical models. However, field measurements of wind speed are typically characterized by large uncertainties due to the generally unknown and uncontrollable boundary conditions (Braham, 1979), the broad range of time and length-scales (Cushman-Roisin and Beckers, 1990a), and the complexity of the physics involved (Stull, 1988). Furthermore, the large measurement volume, which typically extends throughout

the height of the atmospheric boundary layer, imposes to the experimentalists the selection of the sampling parameters as a trade-off between spatial and temporal resolutions.

Wind speed has been traditionally measured through local sensors, such as mechanical, sonic, and hot-wire anemometers (Liu et al., 2019; Kunkel and Marusic, 2006). Besides their simplicity, mechanical anemometers are affected by errors due to the flow distortion of the supporting structures, drawbacks under harsh weather conditions (Mortensen, 1994), and over-

speeding (Busch and Kristensen, 1976). Furthermore, their relatively slow response results in a limited range of the measurable time-length scales, which makes them unsuitable, for instance, to measure the turbulent flow around urban areas (Pardyjak and Stoll, 2017). Sonic anemometers can measure the three velocity components with frequencies up to 100 Hz (Cuerva and Sanz-Andrés, 2000) in a probing volume of the order of $10^{-4}$ m$^3$, yet measurements might be still affected by the wakes generated by the supporting structures, such as met-towers and struts and they are sensitive to temperature variations (Mortensen, 1994).

Hot-wire anemometers, although they provide a full characterization of the energy spectrum, require a complicated calibration (Kunkel and Marusic, 2006) and are extremely fragile (Wheeler and Ganji, 2010a). Furthermore, traditional single-point sensors are unable to provide an adequate characterization of the spatial gradients of the wind velocity vector, which is particularly significant in the vertical direction (Cushman-Roisin and Beckers, 1990b). To overcome this issue, several anemometers arranged in arrays and supported by meteorological masts have been deployed in several field campaigns (Haugen et al., 1971;

Bradley, 1983; Taylor and Teunissen, 1987; Emeis et al., 1995; Pashow et al., 2001; Berg et al., 2011; Kunkel and Marusic, 2006).

In the last few decades, remote sensing instruments have been increasingly utilized to probe the atmospheric boundary layer (Debnath et al., 2017a, b) and nowadays they represent a more cost-effective and flexible alternative to meteorological towers (Newsom et al., 2017). In particular, in the realm of remote sensing anemometry, Doppler wind light detection and ranging

(LiDAR) systems underwent a rapid development due to the significant advancement in eye-safe laser technology (Emeis, 2010). Wind LiDARs have been heavily employed in wind energy (Bingöl et al., 2010; Aitken and Lundquist, 2014; Trujillo et al., 2011; Iungo et al., 2013b; Machefaux et al., 2016; Garcia et al., 2017; El-Asha et al., 2017; Bromm et al., 2018; Zhan et al., 2019, 2020), airport monitoring (Köpp et al., 2005; Tang et al., 2011; Holzäpfel et al., 2016; Thobois et al., 2019), micro-meteorology (Gal-Chen et al., 1992; Banakh et al., 1999; Banta et al., 2006; Mann et al., 2010; Muñoz-Esparza et al.,

2012; Rajewski et al., 2013; Schween et al., 2014), urban wind research (Davies et al., 2007; Newsom et al., 2008; Xia et al., 2008; Kongara et al., 2012; Huang et al., 2017; Halios and Barlow, 2018) and studies of terrain-induced effects (Bingöl, 2009; Krishnamurthy et al., 2013; Kim et al., 2016; Pauscher et al., 2016; Risan et al., 2018; Fernando et al., 2019; Bell et al., 2020).

Besides the mentioned capabilities, LiDARs present some important limitations, such as reduced range in adverse weather conditions (precipitations, heavy rain, fog, low clouds or low aerosol concentration) (Liu et al. (2019), Mann et al. (2018))

and a limited spatio-temporal resolution of this instrument, namely about 20 meters in the radial direction and about 10 Hz in sampling frequency. These technical specifications, associated with the non-stationary wind conditions typically encountered for field experiments, pose major challenges to apply wind LiDARs for the statistical analysis of turbulent atmospheric flows.

In the realm of wind energy, early LiDAR measurements were limited to the qualitative analysis of snapshots of the line-of-sight (LOS) velocity, i.e. the velocity component parallel to the laser beam (Käsler et al., 2010; Clive et al., 2011). Fitting

of the wake velocity deficit was also successfully exploited to extract quantitative information about wake evolution from LiDAR measurements (Aitken and Lundquist, 2014; Wang and Barthelmie, 2015; Kumer et al., 2015; Trujillo et al., 2016; Bodini et al., 2017). To characterize velocity fields with higher statistical significance, the time averages of several LiDAR scans were calculated for periods with reasonably steady inflow conditions (Iungo and Porté-Agel, 2014; Machefaux et al., 2015; Van Dooren et al., 2016). In the case of data collected under different wind and atmospheric conditions, clustering and

bin-averaging of LiDAR data were carried out (Machefaux et al., 2016; Garcia et al., 2017; Bromm et al., 2018; Zhan et al., 2019, 2020). Finally, more advanced techniques for first-order statistical analysis, such as variational methods (Xia et al., 2008; Newsom and Banta, 2004), optimal interpolation (Xu and Gong, 2002; Kongara et al., 2012), least-squares methods (Newsom et al., 2008), Navier-Stokes solvers (Astrup et al., 2017; Sekar et al., 2018) were applied for the reconstruction of the velocity vector field from dual-Doppler measurements.

Besides the mean field, the calculation of higher-order statistics from LiDAR data to investigate atmospheric turbulence is still an open problem. In this regard, Eberhard et al. (1989) re-adapted the post-processing of the velocity azimuth display (VAD) scans (Lhermitte, 1969; Wilson, 1970; Kropfli, 1986) to estimate all the components of the Reynolds stress tensor by assuming horizontal homogeneity of the mean flow within the scanning volume, which can be a limiting constraint for measurements in complex terrains (Frisch, 1991; Bingöl, 2009). Range height indicator (RHI) scans were used to detect second-

order statistics (Bonin et al., 2017), spectra, skewness, dissipation rate of the velocity field, and even heat flux (Gal-Chen et al., 1992). Recently, in the context of wind radar technology, but readily applicable to LiDARs as well, a promising method for the estimation of the instantaneous turbulence intensity (i.e. the ratio between standard deviation and mean of streamwise velocity) based on the Taylor hypothesis of frozen turbulence was proposed by Duncan et al. (2019). More advanced techniques exploit additional information of turbulence carried by the spectrum of the back-scattered LiDAR signal (Smalikho, 1995). However,

this approach requires the availability of LiDAR raw data, which is not generally granted for commercial LiDARs. For a review on turbulence statistical analyses through LiDAR measurements, the reader can refer to Sathe and Mann (2013). Another typical scanning strategy to obtain high-frequency LiDAR data consists in performing scans with fixed elevation and azimuthal angles of the laser beam while maximizing the sampling frequency (Mayor et al., 1997; O'Connor et al., 2010; Vakkari et al., 2015; Frehlich and Cornman, 2002; Debnath et al., 2017b; Choukulkar et al., 2017; Lundquist et al., 2017).

For remote sensing instruments, data are typically collected based on a spherical coordinate system, then interpolated over a Cartesian reference frame oriented with the $x$-axis in the mean wind direction. This interpolation can be a source of error (Fuertes Carbajo and Porté-Agel, 2018), especially if a linear interpolation method is used (Garcia et al., 2017; Carbajo Fuertes

et al., 2018; Beck and Kühn, 2017; Astrup et al., 2017). Delaunay triangulation has also been widely adopted for coordinate transformation (Clive et al., 2011; Trujillo et al., 2011; Iungo and Porté-Agel, 2014; Machefaux et al., 2015; Trujillo et al.,

2016), yet with accuracy not quantified in case of non-uniformly distributed data. It is reasonable to weight the influence of the experimental points on their statistics by the distance from the respective grid centroid, such as using uniform (Newsom et al., 2008), hyperbolic (Van Dooren et al., 2016), or Gaussian weights (Newsom et al., 2014; Wang and Barthelmie, 2015; Zhan et al., 2019). The use of distance-based Gaussian weights for the interpolation of scattered data over a Cartesian grid is at the base of the Barnes objective analysis (or Barnes scheme) (Barnes, 1964), which has been systematically used in meteorology,

but only sporadically for LiDAR data. It represents an iterative statistical ensemble procedure to reconstruct a scalar field arbitrarily sampled in space and low-pass filtered with a cut-off wavelength that is a function of the parameters of the scheme.

The scope of this work is to define a methodology to post-process scattered data of a turbulent velocity field measured through a scanning Doppler wind LiDAR (or eventually other remote sensing instruments) to calculate mean, standard deviation and even higher-order statistical moments on a Cartesian grid. The proposed methodology, referred to as LiDAR Statistical Barnes

Objective Analysis (LiSBOA), represents an adaptation of the classic Barnes scheme to $N$-dimensional domains enabling applications for non-isotropic scalar fields through a coordinate transformation. A major point of novelty of the LiSBOA is the estimation of wind-velocity variance (and eventually higher-order statistics) from the residual field of the mean, which also provides adequate filtering of dispersive stresses due to data variability not connected with the turbulent motion. A criterion for rejection of statistical data affected by aliasing due to the undersampling of the spatial wavelengths under investigation is

formulated. The LiSBOA is assessed against a synthetic scalar field to validate its theoretical response and the formulated error metric. Detailed guidelines for the optimal design of a LiDAR experiment and effective reconstruction of the wind statistics are provided. The effectiveness of the proposed scheme in the identification of the optimal scanning parameters and retrieval of turbulence statistics is quantified using virtual LiDAR data.

It will be shown that the revisited Barnes scheme offers several advantages compared to the above-cited techniques for

LiDAR data analysis: $i$) it allows to explicitly select the cut-off wavenumber to filter out small-scale variability while retaining relevant modes in the flow field; $ii$) the distance-based weighting function provides smoother fields than linear interpolation or window average, while still being simpler and computationally inexpensive compared to more sophisticated techniques (e.g. optimal interpolation, variational methods); $iii$) it provides guidance for the optimal design of LiDAR scans to investigate specific wavelengths in the flow. On the other hand, the procedure requires estimates of input parameters for the flow under

investigation and the LiDAR system used. In case these parameters cannot be obtained from existing literature or preliminary tests, then a sensitivity study on the variability of the LiSBOA results to the input parameters can be carried out.

The remainder of the manuscript is organized as follows: in Sect. 2, the extension of the Barnes scheme theory to $N$-dimensional domains and higher-order statistical moments is presented. In Sect. 3, the theoretical response function of the LiSBOA is validated against a synthetic case, while guidelines for proper use of the proposed algorithm and optimal scan design

are provided in Sect. 4. In Sect. 5, the accuracy of the LiSBOA is tested using of the virtual LiDAR technique. Challenges in the application of the methodology to field experimental data are then discussed in Sect. 6. Finally, concluding remarks are provided in Sect. 7.

## 2   The Barnes Objective Analysis: fundamentals and extension to statistical $N$-dimensional analysis

The Barnes scheme was originally conceived as an iterative algorithm aiming to interpolate a set of sparse data over a Cartesian grid (Barnes, 1964) and it was inspired by the successive correction scheme by Cressman (1959). The first iteration of the algorithm calculates a weighted-space-averaged field, $g^0$, over a Cartesian grid from the sampled scalar field, $f$. The mean field is iteratively modified by adding contributions to recover features characterized by shorter wavelengths, which are inevitably damped by the initial averaging process. In this work, we adopt the most classical form of the Barnes scheme as follows:

$$\begin{cases} g_i^0 = \sum_j w_{ij} f_j \\ g_i^m = \sum_j w_{ij}(f_j - \phi(g^{m-1})_j) + g_i^{m-1} \quad \forall \quad m \in \mathbb{N}^+, \end{cases} \tag{1}$$

where $g_i^m$ is the average field at the $i$-th grid node with coordinates $\boldsymbol{r_i}$ (bold symbols indicate vectorial quantities) for the $m$-th iteration, $f_j$ is the scalar field sampled at the location $\boldsymbol{r_j}$ and $\phi$ represents the linear interpolation operator from the Cartesian grid to the sample location. The weights for the sample acquired at the location $\boldsymbol{r_j}$ and for the calculation of the statistics of $f$ at the grid node with coordinates $\boldsymbol{r_i}$, $w_{ij}$, are defined as:

$$w_{ij} = \frac{e^{-\frac{|\boldsymbol{r_i}-\boldsymbol{r_j}|^2}{2\sigma^2}}}{\sum_j e^{-\frac{|\boldsymbol{r_i}-\boldsymbol{r_j}|^2}{2\sigma^2}}}, \tag{2}$$

where $\sigma$ is referred to as smoothing parameter and $|.|$ indicates Euclidean norm. For practical reasons, the summations over $j$ are performed over the neighboring points included in a ball with a finite radius $R_{\max}$ (also called the radius of influence) and centered at the $i$-th grid point. In this work, following Barnes (1964), we select $R_{\max} = 3\sigma$, which encompasses 99.7% and 97% of the volume of the weighting function in 2D and 3D, respectively.

In literature, there is a lack of consensus for the selection of the total number of iterations (Barnes, 1964; Achtemeier, 1989; Smith and Leslie, 1984; Seaman, 1989) and the smoothing parameter (Barnes, 1994a; Caracena, 1987; Pauley and Wu, 1990). A reduction of the smoothing parameter, $\sigma$, as a function of the iteration, $m$, was originally proposed by Barnes (1973); however, this approach resulted to be detrimental in terms of noise suppression (Barnes, 1994c).

In the frequency domain, the Barnes objective analysis is tractable as a low-pass filter applied to a scalar field, $f$, with a response as a function of the spatial wavelength depending on the smoothing parameter, $\sigma$, and the number of iterations, $m$. This feature has been exploited in meteorology to separate small-scale from mesoscale motions (Doswell, 1977; Maddox, 1980; Gomis and Alonso, 1990). The spectral behavior of the Barnes scheme has been traditionally characterized by calculating the so-called continuous response at the $m$-th iteration, $D^m(\boldsymbol{k})$, with $\boldsymbol{k}$ being the wavenumber vector. $D^m(\boldsymbol{k})$ is defined as the ratio between the amplitude of the Fourier mode $e^{i\boldsymbol{k}\cdot\boldsymbol{x}}$ (with $i = \sqrt{-1}$) for the reconstructed field, $g^m$, to its amplitude for in input field, $f$, in the limit of a continuous distribution of samples and infinite domain. The analytical expression for the continuous response was provided by Barnes (1964) and Pauley and Wu (1990) for 1D and 2D domains, respectively, while in the context of the LiSBOA it is extended to $N$ dimensions to enhance its applicability. Furthermore, besides the spatial variability of $f$, the temporal coordinate, $t$, is introduced to determine the response of the statistical moments of $f$.

We consider a continuous scalar field, $f(\boldsymbol{x}, t)$, which is defined over an $N$-dimensional domain, $\boldsymbol{x}$. It is further assumed that the field $f$ is ergodic in time. In practice, ergodic data can be obtained by selecting samples collected for a temporal window exhibiting stationary boundary conditions or, more generally, through a cluster analysis of discontinuous data (Machefaux et al., 2016; Bromm et al., 2018; Iungo et al., 2018; Zhan et al., 2019, 2020). By adopting the approach proposed by Pauley and Wu (1990) and by taking advantage of the isotropy of the Gaussian weights (Eq. (2)), we can define the LiSBOA operator at the 0-th iteration as:

$$g^0(\boldsymbol{x}) = \frac{1}{(\sqrt{2\pi}\sigma)^N} \int\limits_{\mathbb{R}^N} \left[ \frac{1}{t_2 - t_1} \int\limits_{t_1}^{t_2} f(\boldsymbol{\xi}, t) dt \right] e^{-\frac{|\boldsymbol{x}-\boldsymbol{\xi}|^2}{2\sigma^2}} d\boldsymbol{\xi}, \tag{3}$$

where $t_1$ and $t_2$ are initial and final time. The term within the square brackets represents the mean of $f$ over the considered sampling interval $[t_1, t_2]$, which is indicated as $\overline{f}$. Moreover, to reconstruct a generic $q$-th central statistical moment of the scalar field, $f$, it is sufficient to apply the LiSBOA operator of Eq. (3) to the fluctuations over $\overline{f}$ to the $q$-th power:

$$\mu_f^q(\boldsymbol{x}) = \frac{1}{(\sqrt{2\pi}\sigma)^N} \int\limits_{\mathbb{R}^N} \left\{ \frac{1}{t_2 - t_1} \int\limits_{t_1}^{t_2} \left[ f(\boldsymbol{\xi}, t) - \overline{f}(\boldsymbol{\xi}, t) \right]^q dt \right\} e^{-\frac{|\boldsymbol{x}-\boldsymbol{\xi}|^2}{2\sigma^2}} d\boldsymbol{\xi}. \tag{4}$$

For practical applications, the mean field $\overline{f}$ is generally not known, but it can be approximated by the LiSBOA output, $g^m$, interpolated at the sample location through the operator $\phi$. By comparing Eq. (4) with Eq. (3), it is understandable that the response function of any central moment with an order higher than one is equal to that of the 0-th iteration response of the mean, $g^0$. Indeed, Eq. (3) can be interpreted as the 0-th iteration of the LiSBOA spatial operator (viz. Eq. (3)) applied to the fluctuation field to the $q$-th power.

By leveraging the convolution theorem, it is possible to calculate the response function of the mean of the 0-th iteration of the LiSBOA in the frequency domain (see Appendix A for more details). This result, combined with the recursive formula of Barnes (1964) for the response at the generic iteration $m$, provides the spectral response of the LiSBOA for the mean:

$$D^m = \begin{cases} D^0(\boldsymbol{k}) = e^{-\frac{\sigma^2}{2}|\boldsymbol{k}|^2} = e^{-\frac{\sigma^2\pi^2}{2}\left[\sum_{p=1}^{N} \frac{1}{\Delta n_p^2}\right]} & \text{for } m = 0 \\ D^0 \sum_{p=0}^{m}(1 - D^0)^p & \text{for } m \in \mathbb{N}^+, \end{cases} \tag{5}$$

where $\boldsymbol{\Delta n}$ is the half-wavelength vector associated with $\boldsymbol{k}$. Equation 5 states that, for a given wavenumber (i.e. half-wavelength), the respective amplitude of the interpolated scalar field, $g^m$, is equal to that of the original scalar field, damped with a function of the smoothing parameter, $\sigma$, and the number of iterations, $m$. This implies that the parameters $\sigma$ and $m$ should be selected properly to avoid significant damping for wavelengths of interest or dominating the spatial variability of the scalar field under investigation.

For real applications, the actual LiSBOA response function can depart from the above-mentioned theoretical response (Eq. (5)) for the following reasons:

– the convolution integral in Eq. (3) is calculated over a ball of finite radius $R_{\max}$;

- $f$ is sampled over a discrete domain and, thus, introducing related limitations, such as the risk of aliasing (Pauley and Wu, 1990);

- the distribution of the sampling points is usually irregular and non-uniform leading to larger errors where a lower sample density is present (Smith et al., 1986; Smith and Leslie, 1984; Buzzi et al., 1991; Barnes, 1994a) or in proximity to the domain boundaries (Achtemeier, 1986);

- an error is introduced by the back-interpolation function, $\phi$, from the Cartesian grid, $\boldsymbol{r_i}$, to the location of the samples, $\boldsymbol{r_j}$ (Eq. (1)) (Pauley and Wu, 1990).

Before proceeding with further analysis, it is necessary to address the applicability of the LiSBOA to anisotropic and multichromatic scalar fields. Generally, the application of the LiSBOA with an isotropic weighting function is not recommended in case of severe anisotropy of the field and/or the data distribution. At the early stages of objective analysis techniques, the use of an anisotropic weighting function was proved to be beneficial to increase accuracy while highlighting patterns elongated along a specific direction, based on empirical (Endlich and Mancuso, 1968) and theoretical arguments (Sasaki, 1971). Furthermore, the adoption of a directional smoothing parameter, $\sigma_p$, where $p$ is a generic direction, allows maximizing the utilization of the data retrieved through inherently anisotropic measurements, such as the line-of-sight fields detected by remote sensing instruments (Askelson et al., 2000; Trapp and Doswell, 2000). With this in mind, we propose a linear scaling of the physical coordinates before the application of the LiSBOA to recover a pseudo-isotropic velocity field. The scaling reads as:

$$\tilde{x}_p = \frac{x_p - x_p^*}{\Delta n_{0,p}}, \tag{6}$$

where $\boldsymbol{x^*}$ is the origin of the scaled reference frame and $\Delta n_{0,p}$ is the scaling factor for the $p$-th direction. Hereinafter, $\tilde{\phantom{x}}$ refers to the scaled frame of reference. From a physical standpoint, the scaling is equivalent to the adoption of an anisotropic weighting function, while the re-scaling approach is preferred to ensure generality to the mathematical formulation outlined in this section.

The scaling factor, $\boldsymbol{\Delta n_0}$, is an important parameter in the present framework and is referred to as the fundamental half-wavelength, while the associated Fourier mode is denoted as the fundamental mode. The selection of the fundamental half-wavelength should be guided by a priori knowledge of the dominant length-scales of the flow in various directions. Modes exhibiting degrees of anisotropy different than that of the selected fundamental mode, will not be isotropic in the scaled mapping, which leads to two consequences: first, their response will not be optimal, in the sense that the shortest directional wavelength can produce excessive damping of the specific mode (Askelson et al., 2000); second, the shape preservation of such non-spherical features in the field reconstructed through the LiSBOA is not ensured (Trapp and Doswell, 2000).

Regarding the reconstruction of the flow statistics through the LiSBOA, two categories of error can be identified. The first is the statistical error due to the finite number of samples of the scalar field, $f$, available in time. This error is strictly connected with the local turbulence statistics, the sampling rate, and the duration of the experiment. The second error category is the spatial sampling error, which is due to the discrete sampling of $f$ in the spatial domain $\boldsymbol{x}$. The Petersen-Middleton theorem (Petersen and Middleton, 1962) states that the reconstruction of a continuous and band-limited signal from its samples is possible if and

only if the spacing of the sampling points is small enough to ensure non-overlapping of the spectrum of the signal with his replicas distributed over the so-called reciprocal lattice (or grid). The latter is defined as the Fourier transform of the specific sampling lattice. The 1D version of this theorem is the well-known Shannon-Nyquist theorem (Shannon, 1984). An application of this theorem to non-uniformly distributed samples, like those measured by remote sensing instruments, is unfeasible due to the lack of periodicity of the sampling points. To circumvent this issue, we adopted the approach suggested by Koch et al. (1983), who defined the random data spacing, $\Delta d$, as the equivalent distance that a certain number of samples enclosed in a certain region, $N_{\mathrm{exp}}$, would have if they were uniformly distributed over a structured Cartesian grid. The generalized form of the random data spacing reads:

$$\Delta d(\boldsymbol{r_i}) = \frac{V^{\frac{1}{N}}}{N_{\mathrm{exp}}(\boldsymbol{r_i})^{\frac{1}{N}} - 1}, \tag{7}$$

where $V$ is the volume of the hyper-sphere with radius $R_{\mathrm{max}} = 3\sigma$ centered at the specific grid point and $N_{\mathrm{exp}}$ represents the number of not co-located sample locations included within the hyper-sphere. Then, the Petersen-Middleton theorem for the reconstruction of the generic Fourier mode of half-wavelength $\boldsymbol{\Delta n}$ can be translated as the following constraint:

$$\Delta d(\boldsymbol{r_i}) < \Delta n_p, \ p = 1, 2, ..., N. \tag{8}$$

Violation of the inequality (8) will lead to local aliasing, with the energy content of the under-sampled wavelengths being added to the low-frequency part of the spectrum.

## 3    LiSBOA assessment through Monte Carlo simulations

The spectral response of the LiSBOA is studied through the Monte Carlo method. The goal of the present section is twofold: validating the analytical response of mean and variance (Eq. (5)) and characterizing the sampling error of the LiSBOA as a function of the random data spacing. For these aims, a synthetic 3D scalar field is generated, while its temporal variability is reproduced locally by randomly sampling a normal probability density function. Specifically, the synthetic scalar field is:

$$f = \left[1 + \sin\left(\frac{\pi}{\Delta n}x\right)\sin\left(\frac{\pi}{\Delta n}y\right)\sin\left(\frac{\pi}{\Delta n}z\right)\right] + \left[1 + \sin\left(\frac{\pi}{\Delta n}x\right)\sin\left(\frac{\pi}{\Delta n}y\right)\sin\left(\frac{\pi}{\Delta n}z\right)\right]^{0.5} \aleph(0,1), \tag{9}$$

where $\aleph$ is a generator of random numbers with normal probability density function with mean value $0$ and standard deviation equal to $1$. The constant $1$ in the two terms on the RHS of Eq. (9) does not affect the LiSBOA response and is introduced to obtain both mean and variance of $f$ equal to the following function:

$$\overline{f} = 1 + \sin\left(\frac{\pi}{\Delta n}x\right)\sin\left(\frac{\pi}{\Delta n}y\right)\sin\left(\frac{\pi}{\Delta n}z\right). \tag{10}$$

It is noteworthy that $\overline{f} - 1$ is a monochromatic isotropic function.

An experimental sampling process is mimicked by evaluating the scalar field $f$ through randomly and uniformly distributed samples collected at the locations $\boldsymbol{r_j}$. The latter are distributed within a cube spanning the range $\pm 10\sigma$ in the three Cartesian directions. The total number of sampling points considered for each realization, $N_s$, is varied from 500 up to 20,000 to explore

the effects of the sample density on the error. The sampling process is repeated $L$ times for each given distribution of $N_s$ points to capture the variability in the field introduced by the operator $\aleph$. The whole procedure can be considered as an idealized

LiDAR experiment where a scan including $N_s$ sampling points is performed $L$ times to probe an ergodic turbulent velocity field.

Since the response is only a function of $\Delta n/\sigma$ and $m$ (Eq. (5)), for the spectral characterization of the LiSBOA, the parameter $\Delta n/\sigma$ is varied among the following values: [1, 2, 3, 4, 5]. An implementation of the LiSBOA algorithm for discrete samples is then applied to reconstruct the mean $g^m$ and variance $v^m$ of the scalar field $f$ over a Cartesian structured grid, $r_i$, with a

275 resolution of 0.25. Figure 1 depicts an example of the reconstruction of the mean scalar field, $g^m$, and its variance, $v^m$, from the Monte Carlo synthetic dataset.

For the error quantification, the 95-th percentile of the absolute error calculated at each grid point $r_i$ ($AE_{95}$ hereinafter) is adopted:

$$AE_{95}(\Delta n/\sigma, m, N_s, L) = \begin{cases} \text{percentile}_{95}\langle|(g^m-1)-D^m(\overline{f}-1)|\rangle_{r_i} & \text{for the mean} \\ \text{percentile}_{95}\langle|(v^m-1)-D^0(\overline{f}-1)|\rangle_{r_i} & \text{for the variance.} \end{cases} \tag{11}$$

The $AE_{95}$ quantifies the discrepancy between the outcome of the LiSBOA and the analytical input damped by the theoretical response evaluated over the Cartesian grid. As highlighted in Eq. (11), the expected value of $AE_{95}$ is a function of the half-wavelength over the smoothing parameter, $\Delta n/\sigma$, the number of iterations, $m$, the number of samples, $N_s$, and the number of realizations, $L$. To investigate the link between $AE_{95}$ and the above-mentioned parameters, the Pearson correlation coefficients are analyzed (Table 1). The number of samples $N_s$, which is inversely proportional to the data spacing $\Delta d$ (Eq. 7), is the

variable exhibiting the strongest correlation with the error for both mean and variance. This indicates, as expected, that a larger number of samples for each measurement realization is always beneficial for the estimates of the statistics of the scalar field, $f$. Furthermore, the negative sign of correlations $\rho(AE_{95}, N_s)$ and $\rho(AE_{95}, \Delta n/\sigma)$, corroborate the hypothesis that the ratio

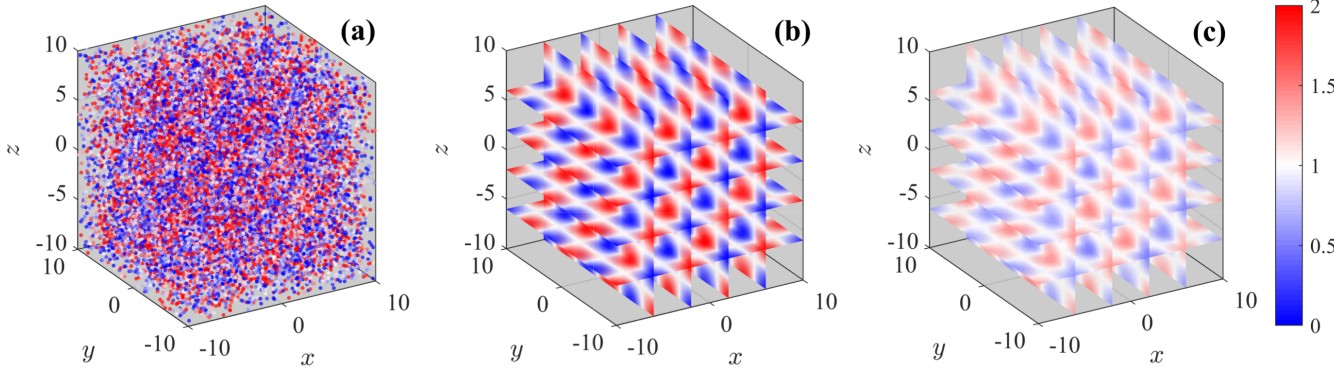

**Figure 1.** Visualization of the LiSBOA applied to a Monte Carlo simulation of the synthetic field in Eq. (9) for the case with $N_s = 20,000$, $L = 200$, $\Delta n/\sigma = 4$ and $m = 5$: **(a)** samples; **(b)** 3D reconstructed mean field, $g^m$; **(c)** 3D reconstructed variance, $v^m$.

$\Delta d / \Delta n$, i.e. the number of samples per half-wavelength, is the main driving factor for the sampling error (Koch et al., 1983; Barnes, 1994a; Caracena et al., 1984).

The small positive correlation $\rho(AE_{95}, m)$ detected for the mean is due to an amplification of the error occurring during the iterative process (Barnes, 1964). The issue will be discussed more in detail in Sect. 4. For the variance, $\rho(AE_{95}, m)$ is practically negligible, confirming that the response of the higher-order statistics is insensitive to the number of iterations, $m$. Finally, the negative correlations with $L$ show that the statistical error is inversely proportional to the number of realizations collected. The dependence $\rho(AE_{95}, L)$ is mainly due to the statistical error connected with the temporal sampling and, thus, the

number of realizations, $L$, is progressively increased until convergence of the $AE_{95}$ is achieved. Figure 2 displays the behavior of the error as a function of $N_s$ and $L$. The values displayed represent the median for all the wavelength and iterations, being the $AE_{95}$ just mildly dependent on these parameters. As Fig. 2 shows, increasing the number of realizations, $L$, beyond 100 has a negligible effect on the error, thus a final value of $L = 200$ is selected for the remainder of this analysis.

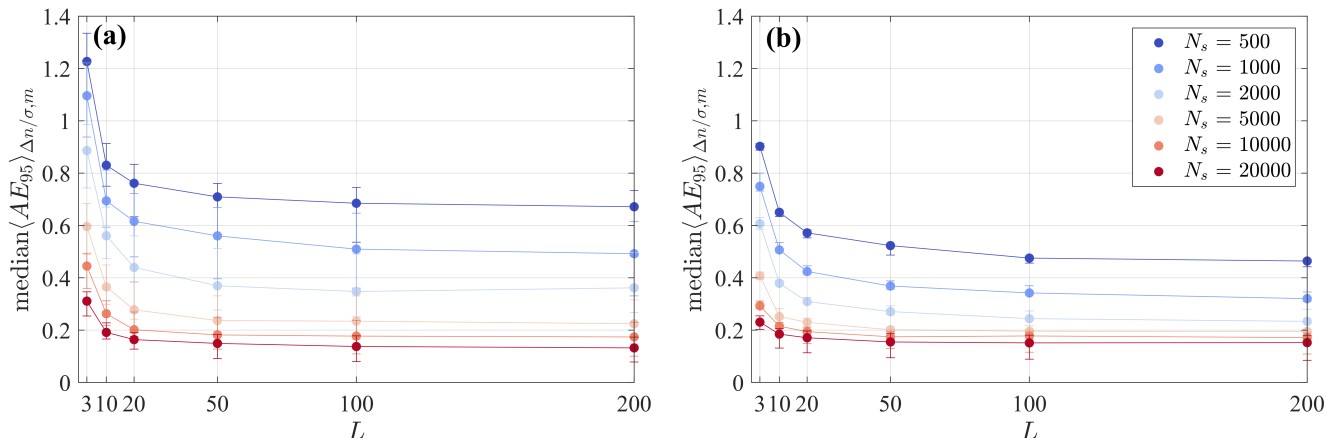

**Figure 2.** Median of the $AE_{95}$ for all the tested half-wavelengths, $\Delta n / \sigma$, and the number of iterations, $m$: **(a)** $AE_{95}$ of the mean field, $g^m$; **(b)** $AE_{95}$ of the variance field, $v^m$. The error bars span the interquartile range.

To verify the analytical response of mean and variance of the scalar field, $f$, a numerical estimator of the response is defined as the median in space of the ratio between the field reconstructed via LiSBOA and the expected value of the synthetic input,

**Table 1.** Pearson correlation coefficient between the $AE_{95}$ of mean and variance and the parameters $\Delta n / \sigma$, $m$, $N_s$, $L$. The values between parenthesis represent the 95% confidence bounds.

|  | $\Delta n / \sigma$ | $m$ | $N_s$ | $L$ |
|---|---|---|---|---|
| $AE_{95}$ of mean | -0.259 (-0.303, -0.210) | 0.257 (0.211, 0.301) | -0.709 (-0.732, -0.684) | -0.171 (-0.217, -0.124) |
| $AE_{95}$ of variance | -0.069 (-0.117, -0.021) | -0.03 (-0.078, 0.019) | -0.694 (-0.718, -0.668) | -0.206 (-0.251, -0.159) |

as:

$$\begin{cases} D^m = \mathrm{median}\langle\frac{g^m-1}{\overline{f}-1}\rangle_{\boldsymbol{r_i}} & \text{for the mean} \\ D^0 = \mathrm{median}\langle\frac{v^m-1}{\overline{f}-1}\rangle_{\boldsymbol{r_i}} & \text{for the variance.} \end{cases} \tag{12}$$

In the calculation of the numerical response through Eq. (12), the influence of the edges is removed by rejecting points closer than $R_{\mathrm{max}}$ to the boundaries of the numerical domain. Furthermore, the zero-crossings of the synthetic sine function ($|\overline{f}-1| <$ 0.1) are excluded to avoid singularities. A comparison between the actual and the theoretical response (Eq. (5)) for several wavelengths of the input function is reported in Fig. 3 for the case with the highest number of samples $N_s = 20,000$. An excellent agreement is observed between the theoretical prediction and the Monte Carlo outcome, which indicates that in the limit of negligible statistical error (large $L$) and adequate sampling (large $N_s$ and near-uniformly distributed samples) the response approaches the predictions obtained from the developed theoretical framework.

The trend of the response of the mean (Fig. 3a) suggests that, for a given wavelength, the same response can be achieved for an infinite number of combinations $\sigma - m$ and, specifically, a larger $\sigma$ requires a larger number of iterations, $m$, to achieve a certain response $D^m$. It is noteworthy that for a smaller number of iterations, $m$, the slope of the response function is lower. This feature can be beneficial for practical applications for which the LiSBOA response will have small changes for small variations of $\Delta n$. However, a lower slope of the response function can be disadvantageous for short-wavelength noise suppression. Figure 3b confirms that the response of the variance, and similarly for higher-order statistics, is not a function of the total number of iterations, $m$, and is equal to the response of the mean for the 0-th iteration, $D^0$.

Finally, the link between error and the random data spacing, $\Delta d$, is investigated. In Fig. 4, the discrepancy with respect to theory quantified by the $AE_{95}$ is plotted versus the random data spacing normalized by the half-wavelength for a fixed total number of iterations $m = 5$. The values displayed on the x-axis represent the median over all grid points, $\boldsymbol{r_i}$. This analysis reveals a strong correlation between the normalized random data spacing and the error. This analysis corroborates that, in the

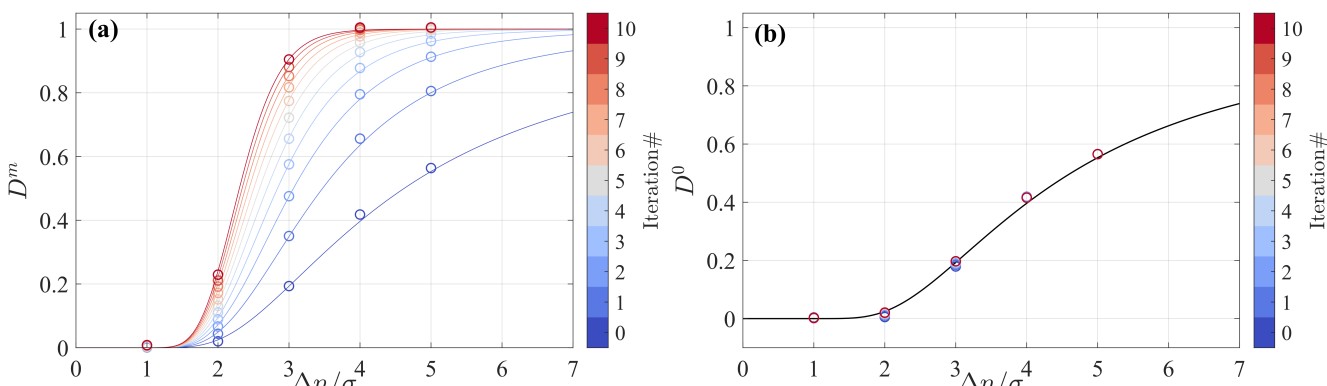

**Figure 3.** Validation of the 3D theoretical response of the LiSBOA for the case $N_s = 20000$ - $L = 200$: **(a)** mean; **(b)** variance. The circles are the numerical output of the Monte Carlo simulation (Eq. (12)), while the continuous lines represent Eq. (5).

limit of negligible statistical error (viz. a high number of realizations, $L$), uncertainty is mainly driven by the local data density normalized by the wavelength, which is related to the Petersen-Middleton criterion. Indeed, the cases satisfying the Petersen-Middleton constraint (Eq. (8)) are those exhibiting an $AE_{95}$ smaller than $\sim 40\%$ of the amplitude of the harmonic function $\overline{f}$ for both mean and variance. However, if a smaller error is needed, it will be necessary to reduce the maximum threshold value for $\Delta d/\Delta n$.

## 4   Guidelines for an efficient application of the LiSBOA to wind LiDAR data

An efficient application of the LiSBOA to LiDAR data relies on the appropriate selection of the parameters of the algorithm, namely the fundamental half-wavelengths, $\boldsymbol{\Delta n_0}$, the smoothing parameter, $\sigma$, the number of iterations, $m$, and the spatial discretization of the Cartesian grid, $\boldsymbol{dx}$. Furthermore, the data collection strategy must be designed to ensure adequate sampling of the spatial wavelengths of interest, so that the Petersen-Middleton constraint (Eq. (8)) is satisfied. In this section, we show that the underpinning theory of the LiSBOA, along with an estimate of the properties of the flow under investigation, can guide the optimal design of a LiDAR experiment and evaluation of the statistics for a turbulent ergodic flow. The whole procedure can be divided into three phases: characterization of the flow, design of the experiment, and reconstruction of the statistics from the collected dataset.

Firstly, the integral quantities of the flow under investigation required for the application of the LiSBOA need to be estimated, such as extension of the spatial domain of interest, characteristic length-scales, integral time-scale, $\tau$, characteristic temporal variance of the velocity, $\overline{u'^2}$, and expected total sampling time, $T$, which depends on the typical duration of stationary boundary conditions over the domain. These estimates can be based on previous studies available in the literature, numerical simulations, or preliminary measurements.

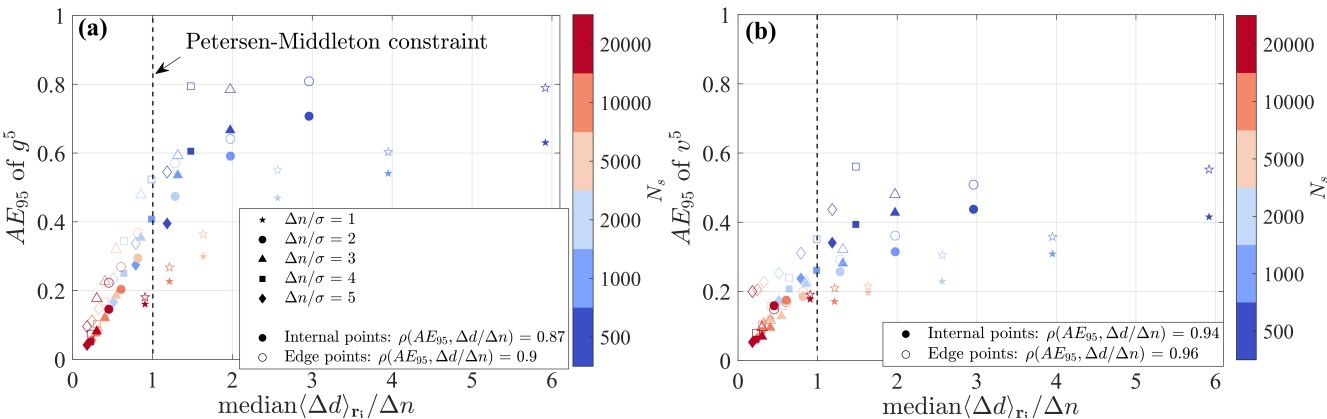

**Figure 4.** $AE_{95}$ as a function of the random data spacing (Eq. (7)) for the case with $m = 5$, $N = 20000$ and $L = 200$: **(a)** error on the mean; **(b)** error on the variance. The full symbols refer to points not affected by the presence of the finite boundaries of the domain, while the empty symbols are taken within a distance of less than $R_{\text{max}}$ from the boundaries.

**Table 2.** Selected combinations of $\sigma$ and $m$ to achieve a $\sim95\%$ recovery of the mean of the selected fundamental half-wavelength and associated response of the higher-order moments (HOM).

| | $N = 2$ | | | | $N = 3$ | | |
|---|---|---|---|---|---|---|---|
| $\sigma$ | $m$ | $D^m$ (mean) | $D^0$ HOM | $\sigma$ | $m$ | $D^m$ (mean) | $D^0$ (HOM) |
| 1/3 | 6 | 0.942 | 0.334 | 1/4 | 5 | 0.952 | 0.397 |
| 1/4 | 3 | 0.955 | 0.540 | 1/6 | 2 | 0.961 | 0.663 |
| 1/6 | 1 | 0.942 | 0.76 | 1/8 | 1 | 0.957 | 0.793 |
| 1/13 | 0 | 0.943 | 0.943 | 1/17 | 0 | 0.950 | 0.950 |

Then, it is necessary to define the fundamental half-wavelengths, $\boldsymbol{\Delta n_0}$, which are required for the coordinate scaling (Eq. (6)). It is advisable to impose the fundamental half-wavelengths equal to (or even smaller than) the estimated characteristic length-scales of the smallest spatial features of interest in the flow. This ensures isotropy of the mode associated with the fundamental half-wavelength (and all the modes characterized by the same degree of anisotropy) and guides the selection of the main input parameters of the LiSBOA algorithm, i.e. smoothing parameter, $\sigma$, and number of iterations, $m$. Indeed, $\boldsymbol{\Delta n_0}$ can be considered as the cut-off half-wavelength of the spatial low-pass filter represented by the LiSBOA operator. To this aim, it is necessary to select $\sigma$ and $m$ to obtain a response of the mean associated with the fundamental mode, $D^m(\boldsymbol{\Delta \tilde{n}_0})$, as close as possible to 1. After the coordinate scaling (Eq. (6)), the response of the fundamental mode is universal and it is reported in Fig. 5. For instance, if we select a response equal to 0.95, then all the points lying on the iso-contour defined by the equality $D^m(\boldsymbol{\Delta \tilde{n}_0}) = 0.95$ give, in theory, the same response for the mean of the scalar field $f$. This implies that an infinite number of combinations $\sigma - m$ allow obtaining a response of the mean equal to the selected value. However, with increasing $\sigma$ the response at the 0-th iteration, $D^0(\boldsymbol{\Delta \tilde{n}_0})$, reduces, which indicates a lower response for higher-order statistics. For the LiSBOA application, the following aspects should be also considered:

– the smaller $\sigma$, the smaller the radius of influence of the LiSBOA, $R_{\max}$, and, thus, the lower the number of samples averaged per grid node, $N_{\exp}$, and the greater the statistical uncertainty;

– an excessively large $m$ can lead to overfitting of the experimental data and noise amplification (Barnes, 1964);

– the higher $m$, the higher the slope of the response function (see Fig. 3), which improves the damping of high-frequency noise, but it produces a larger variation of the response of the mean with different spatial wavelengths;

– the radius of influence $R_{\max}$ (and therefore $\sigma$) can affect the data spacing $\Delta d$ in case of non-uniform data distribution.

Few handy combinations of smoothing parameter and total iterations for $D^m(\boldsymbol{\Delta \tilde{n}_0}) = 0.95$ are provided in Table 2. As mentioned above, all these $\sigma - m$ pairs allow achieving roughly the same response for the mean, while the response for the higher-order statistics reduces with an increasing number of iterations, $m$.

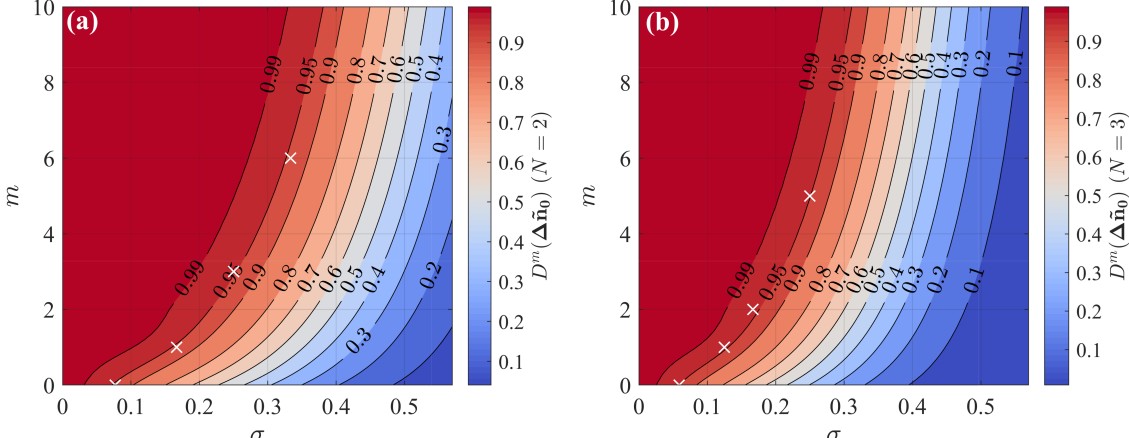

**Figure 5.** Response of the fundamental mode in the scaled coordinates as a function of the number of iterations and the smoothing parameter: **(a)** two-dimensional LiSBOA; **(b)** three-dimensional LiSBOA. The white crosses indicate the pairs $\sigma - m$ provided in Table 2.

As a final remark about the selection of $\mathbf{\Delta n_0}$, we should consider that, if the fundamental half-wavelength is too large compared to the dominant modes in the flow, small-scale spatial oscillations of $f$ will be smoothed out during the calculation of the mean, with consequent underestimated gradients and incorrect estimates of the high-order statistics due to the dispersive 365  stresses (Arenas et al., 2019). On the other hand, the selection of an overly small $\mathbf{\Delta n_0}$, would require an excessively fine data spacing to satisfy the Petersen-Middleton constraint (Eq. (8)), which may lead to an overly long sampling time or even exceed the sampling capabilities of the LiDAR.

The optimal LiDAR scanning strategy aimed to characterize atmospheric turbulent flows implies finding a trade-off between a sufficiently fine data spacing, which is quantified through $\Delta d$ in the present work (Eq.(7)), and an adequate number of 370  time realizations, $L$, to reduce the temporal statistical uncertainty. Considering a total sampling period $T$, for which statistical stationarity can be assumed, and a pulsed LiDAR that scans $N_r$ points evenly spaced along the LiDAR laser beam, with a range gate $\Delta r$ and accumulation time $\tau_a$, the total number of collected velocity samples is then equal to $N_s = N_r \cdot T/\tau_a$. The angular resolution of the LiDAR scanning head in azimuth ($\Delta\theta$, for PPIs) or elevation ($\Delta\beta$, for RHIs) or both axis (for volumetric scans), can be selected to modify the angular spacing between consecutive line-of-sights (i.e. the data spacing) and the total 375  sampling period for a single scan, $\tau_s$ (i.e. the number of realizations, $L$).

The design of a LiDAR scan aiming to reconstruct turbulent statistics of an ergodic flow through LiSBOA can be formalized as a two-objective (or Pareto front) optimization problem. The first cost function of the Pareto front, which is referred to as $\epsilon^I$, is the percentage of grid nodes for which the Petersen-Middleton constraint applied to the smallest half-wavelength of interest (i.e. $\mathbf{\Delta n_0}$), is not satisfied. In respect to the scaled reference frame, this can be expressed as:

$$380 \quad \epsilon^I(\Delta\theta, \Delta\beta, \sigma) = \frac{\sum_{i=1}^{N_i}[\Delta\tilde{d} > 1]}{N_i}, \tag{13}$$

where the square brackets are Iverson brackets and $N_i$ is the total number of nodes in the Cartesian grid, $\boldsymbol{r_i}$. For a more conservative formulation, it is recommended to reject all the points with a distance smaller than $R_{\max}$ from an under-sampled grid node, i.e. with $\Delta\tilde{d} > 1$. This condition will ensure that the statistics are based solely on regions that are adequately sampled. The cost function $\epsilon^I$ depends not only on the angular resolution but also on $R_{\max}$, which is equal to $3\sigma$ in this work. In general, increasing $\sigma$ results in a larger number of samples considered for the calculation of the statistics at each grid point $\boldsymbol{r_i}$ and, thus, in a reduction of $\epsilon^I$. Therefore, a larger $\sigma$ entails a larger percentage of the spatial domain fulfilling the Petersen-Middleton constraint. The smoothing parameter, $\sigma$, also plays a fundamental role in the response of higher-order statistical moments. Specifically, if the reconstruction of the variance or higher-order statistics is important, the response $D^0(\boldsymbol{\Delta\tilde{n}_0})$ should be included in the Pareto front analysis as an additional constraint.

The second cost function for the optimal design of LiDAR scans, $\epsilon^{II}$, is equal to the standard deviation of the sample mean, which, for an autocorrelated signal, is (Bayley and Hammersley, 1946):

$$\epsilon^{II}(\Delta\theta, \Delta\beta) = \sqrt{\overline{u'^2}}\sqrt{\frac{1}{L} + \frac{2}{L^2}\sum_{p=1}^{L-1}(L-p)\,\rho_p} \sim \sqrt{\overline{u'^2}}\sqrt{\frac{1}{L} + \frac{2}{L^2}\sum_{p=1}^{L-1}(L-p)\,e^{-\frac{\tau_s}{\tau}p}}, \tag{14}$$

where $\rho_p$ is the autocorrelation function at lag $p$, $\tau$ is the integral time-scale and the approximation is based on George et al. (1978). The velocity variance, $\overline{u'^2}$, and the autocorrelation, $\rho_p$, are functions of space; however, to a good degree of approximation, they can be replaced by a representative value and considered as uniform in space. Figure 6 shows the standard deviation of the sample mean normalized by the standard deviation of the velocity as a function of the number of realizations, $L$, and for different integral time-scales, $\tau$. It is noteworthy that the standard deviation of the sample mean represents the uncertainty of the time-average of each measurement point, $\boldsymbol{r_j}$, while the final uncertainty of the mean field at the grid nodes $\boldsymbol{r_i}$ is generally reduced due to the spatial averaging process intrinsic to the LiSBOA. It is noteworthy that the estimates of the statistical error obtained through the LiSBOA do not consider other sources of error, such as accuracy of the instruments and spatial averaging due to the LiDAR measuring process (Rye and Hardesty, 1993; O'Connor et al., 2010; Puccioni and Iungo, 2020). Eventually, other error estimates can be coupled with the sampling error estimated through the LiSBOA for a more comprehensive error analysis (Wheeler and Ganji, 2010b). Furthermore, LiSBOA allows calculating velocity statistics including contributions of eddies with different sizes, which span from the largest eddy advected within the total sampling time, to the smallest eddy detectable for a given accumulation time (Puccioni and Iungo, 2020). Therefore, a careful pre-processing of the LiDAR data should be eventually performed to remove contributions due to non-turbulent mesoscale eddies (Högström et al., 2002; Metzger et al., 2007; O'Connor et al., 2010).

The whole procedure for the design of a LiDAR scan and retrieval of the statistics is reported in the flow chart of Fig. 7. Summarizing, from a preliminary analysis of the velocity field under investigation, we estimate the maximum total sampling time, $T$, the characteristic integral time-scale, $\tau$, the characteristic velocity variance, $\overline{u'^2}$, the fundamental half-wavelengths $\boldsymbol{\Delta n_0}$. This information, together with the settings of the LiDAR (namely, the accumulation time, $\tau_a$, the number of points per beam, $N_r$, and the gate length, $\Delta r$), allow for generating the Pareto front as a function of $\Delta\theta$ and/or $\Delta\beta$, and for different values of $\sigma$. Based on the specific goals of the LiDAR campaign in terms of coverage of the selected domain (i.e. $\epsilon^I$), statistical

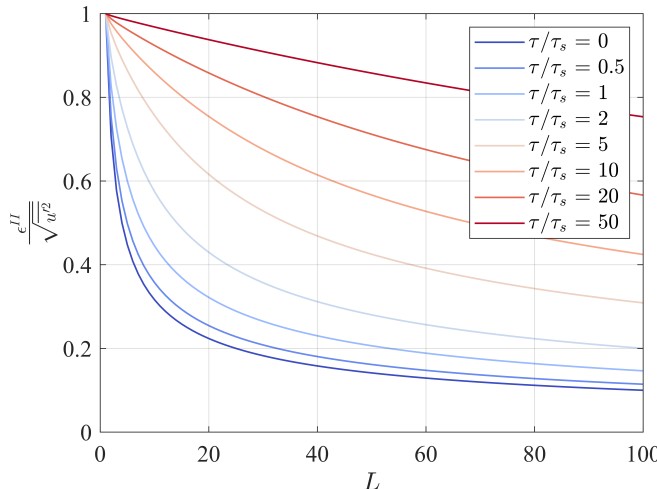

**Figure 6.** Standard deviation of the sample mean normalized by the standard deviation of velocity as a function of the number of realizations, $L$, and for different values of the ratio between the integral time-scale and the sampling time, $\tau/\tau_s$.

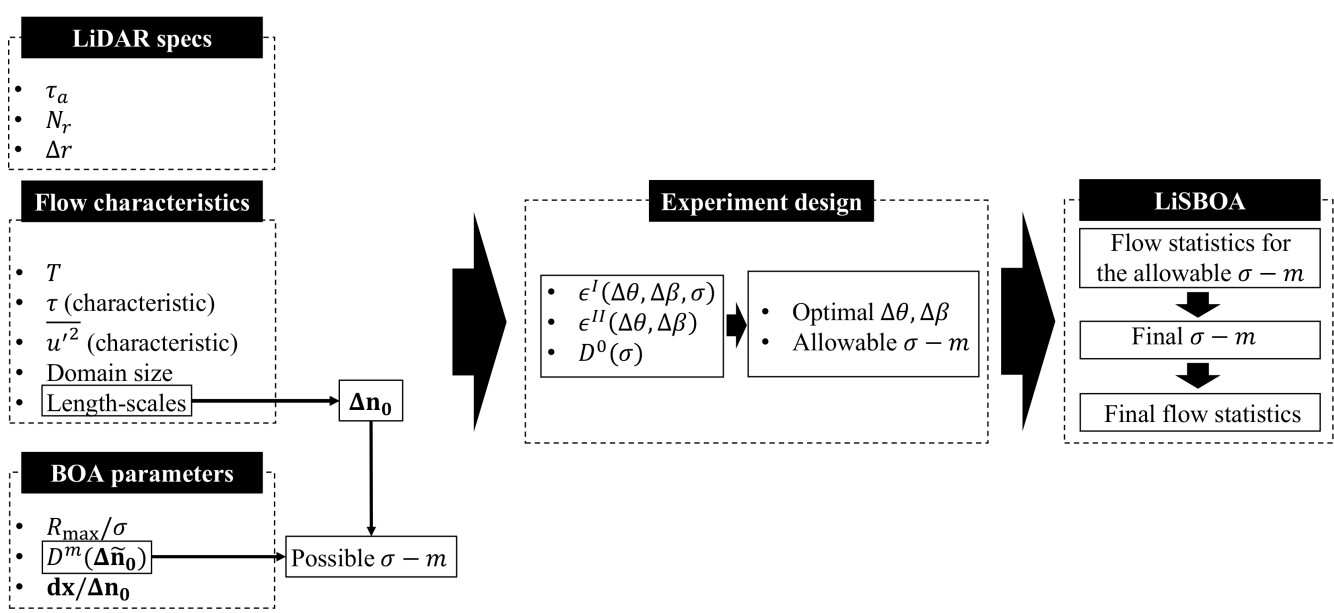

**Figure 7.** Schematic of the LiSBOA procedure for the optimal design of LiDAR scans and reconstruction of the statistics for a turbulent ergodic flow.

significance of the data (i.e. $\epsilon^{II}$) and, eventually, response of the higher-order statistical moments (i.e. $D^0(\boldsymbol{\Delta\tilde{n}_0})$), the LiSBOA user should select the optimal angular resolution, $\Delta\theta$ and/or $\Delta\beta$, and the set of allowable $\sigma$ values. Due to the above-mentioned non-ideal effects on the LiSBOA, the selection of $\sigma$ is finalized during the post-processing phase when the LiDAR dataset is

available and the statistics can be calculated for different pairs of $\sigma - m$ values. For the resolution of the Cartesian grid, Koch et al. (1983) suggested that it should be chosen as a fraction of the data spacing, which, in turn, is linked to the fundamental half-wavelength. The same author suggested a grid spacing included in the range $\boldsymbol{dx} \in [\boldsymbol{\Delta n_0}/3, \boldsymbol{\Delta n_0}/2]$. In this work, we have used $\boldsymbol{dx} = \boldsymbol{\Delta n_0}/4$, which ensures a good grid resolution with acceptable computational costs.

By following the steps outlined in the present section, the mean, variance, or even higher-order statistical moments of the velocity field can be accurately reconstructed for the wavelengths of interest. It is worth mentioning that the LiSBOA of wind LiDAR data should always be combined with a robust quality-control process of the raw measurements. Indeed, the space-time averaging operated by the LiSBOA makes the data analysis sensitive to the presence of data outliers, which need to be identified and rejected beforehand to prevent contamination of the final statistics. The interested reader is referred to Manninen et al. (2016); Beck and Kühn (2017); Vakkari et al. (2019) for more information on quality control of LiDAR data. On a final note, for applications of the LiSBOA, the uncontrollable environmental conditions and the uncertainty in the flow characteristics needed as the input of the LiSBOA may pose some challenges, which will be discussed more in detail in Sect. 6.

## 5 LiSBOA validation against virtual LiDAR data

The LiSBOA algorithm is applied to a synthetic dataset generated through the virtual LiDAR technique to assess accuracy in the calculation of statistics for a wind turbine wake probed through a scanning LiDAR installed at the turbine nacelle. For this purpose, a simulator of a scanning Doppler pulsed wind LiDAR is implemented to extract the line-of-sight velocity from a numerical velocity field produced through high-fidelity large-eddy simulations (LES). Due to their simplicity and low computational costs, LiDAR simulators have been widely used for the assessment of post-processing algorithms of LiDAR data and scan design procedures (Mann et al., 2010; Stawiarski et al., 2015; Lundquist et al., 2015; Mirocha et al., 2015).

As a case study, we use the LES dataset of the flow past a single turbine with the same characteristics of the 5-MW NREL reference wind turbine (Jonkman et al., 2009). The rotor is three-bladed and has a diameter $D = 126$ m. The tip-speed ratio of the turbine is set to its optimal value of 7.5. A uniform incoming wind with freestream velocity of $U_\infty = 10$ m s$^{-1}$ and turbulence intensity of $3.6\%$ is considered. The rotor is simulated through an actuator disk with rotation, while the drag of the nacelle is taken into account using an immersed boundary method (Ciri et al., 2017). More details on the LES solver can be found in Santoni et al. (2015). The computational domain has dimensions ($L_x \times L_y \times L_z = 12D \times 6D \times 6D$) in the streamwise, spanwise, and vertical directions, respectively, and it is discretized with $960 \times 256 \times 300$ uniformly spaced grid points, respectively, resulting in a spacing of $dx = 0.0125D$, $dy = 0.025D$ and $dz = 0.0202D$. A radiative condition is imposed at the outlet (Orlanski, 1976), while periodicity is applied in the spanwise direction. For the sake of generality, a uniform incoming wind is generated by imposing freeslip conditions at the top and bottom of the numerical domain. Ergodic velocity vector fields are available for a total time of $T = 750$ s.

For the estimation of the flow characteristics necessary for the scan design, the azimuthally-averaged mean and standard deviation of streamwise velocity, as well as the integral time-scale are considered (Fig. 8). The use of cylindrical coordinates is

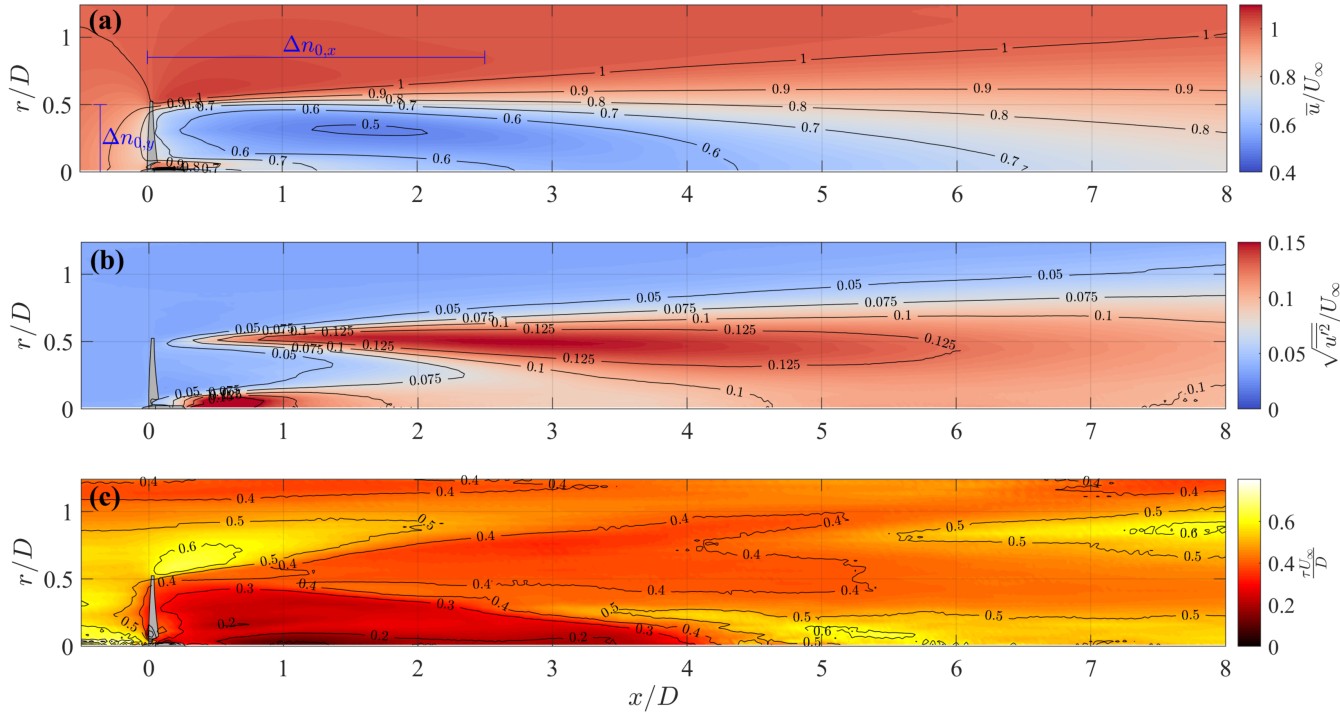

**Figure 8.** Azimuthally-averaged statistics of the LES streamwise velocity field: **(a)** mean value; **(b)** standard deviation; **(c)** integral time scale.

justified by the axisymmetry of the statistics of the wake velocity field generated by a turbine operating in a uniform velocity field (Iungo et al., 2013a; Viola et al., 2014; Ashton et al., 2016).

The streamwise LES velocity field shows the presence of a higher-velocity jet surrounding the nacelle, while $\overline{u}/U_\infty$ exhibits a clear minimum placed at $y/D \sim 0.25$ (Fig. 8a). These flow features are consistent with the double-Gaussian velocity profile typically observed in the near-wake region (Aitken and Lundquist, 2014). In Fig. 8b, the standard deviation of the streamwise velocity has high values in the very near-wake ($x/D < 1$) in the proximity of the rotor axis, which is most probably connected with the vorticity structures generated in proximity of the rotor hub and their dynamics (Iungo et al., 2013a; Viola et al., 2014; Ashton et al., 2016). Similarly, enhanced values of the velocity standard deviation occur at the wake boundary ($r/D \approx 0.5$), which are connected with the formation and dynamics of the helicoidal tip vortices (Ivanell et al., 2010; Debnath et al., 2017c). A peak of $\sqrt{\overline{u'^2}}/U_\infty$ is observed around ($x/D \approx 3$), which can be considered as the formation length of the tip vortices. The integral time-scale is evaluated integrating the sample biased autocorrelation function of the time series of $u$ up to the first zero-crossing (Zieba and Ramza, 2011). The integral time-scale is generally smaller within the wake than for the typical values observed in the freestream, which is consistent with the smaller dimensions of the wake vorticity structures compared to the larger energy-containing structures present in the incoming turbulent wind.

To reconstruct the mentioned flow features, the fundamental half-wavelengths in the spanwise and vertical directions selected for this application of the LiSBOA are $\Delta n_{0,y} = \Delta n_{0,z} = 0.5D$, which allows retrieving spatial features of the velocity field as small as the rotor blade in the cross-stream direction, which are typically observed in the near wake (Aitken and Lundquist, 2014; Santoni et al., 2017). Furthermore, considering the streamwise elongation of the isocontours of the flow statistics shown in Fig. 8a, a conservative value of the fundamental half-wavelength in the $x$-direction $\Delta n_{0,x} = 2.5D$ is selected. This information could have been inferred also from previous field measurements (e.g. Chamorro and Porté-Agel (2010); Abkar and Porté-Agel (2013); Zhan et al. (2019)).

The availability of the LES dataset allows testing the relevance of the selected $\boldsymbol{\Delta n_0}$ by evaluating the 3D energy spectrum of $\overline{u}/U_\infty$ and $\overline{u'^2}/U_\infty^2$ in the physical and in the scaled reference frame (Eq. (6)). The spectra are azimuthally averaged by exploiting the axisymmetry of the wake. The spectra in the physical reference frame (Figs. 9a and b) reveal the clear signature of a streamwise elongation of the energy-containing scales for both velocity mean and variance, with the energy being spread

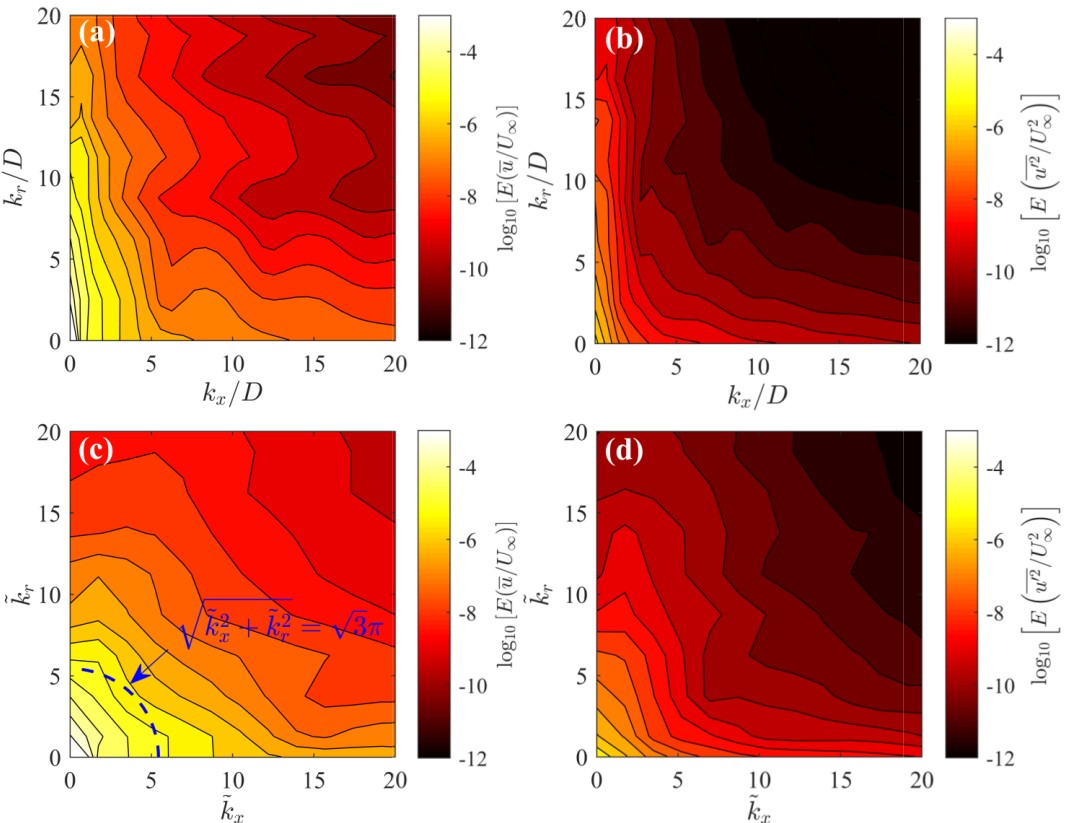

**Figure 9.** Azimuthally-averaged energy spectra of the LES velocity fields : **(a)** mean streamwise velocity on the physical domain; **(b)** variance of streamwise velocity on the physical domain; **(c)** mean streamwise velocity on the scaled domain; **(d)** variance of streamwise velocity on the scaled domain. The blue dashed line indicates wavenumbers reconstructed with response equal to $D^m(\boldsymbol{\Delta \tilde{n}_0}) = 0.95$.

over a larger range of frequencies in the radial direction compared to the streamwise direction. After the scaling (Figs. 9c and d), the spectra become more isotropic in the spectral domain, namely the energy is distributed equally along the $\tilde{k}_x$ and $\tilde{k}_r$ axes. In Fig. 9c, the blue dashed line represents the intersection with the $\tilde{k}_x$ - $\tilde{k}_r$ plane of the spherical isosurface that in the wavenumber space is characterized by $D^m(\mathbf{\Delta \tilde{n}_0}) = 0.95$. All the modes contained within that sphere are reconstructed with a response $D^m > 0.95$, while higher-frequency features lying outside will be damped. Numerical integration of the 3D energy spectrum shows that 94% of the total spatial variance of the mean is contained within that sphere, which ensures that the energy-containing modes in the mean flow are adequately reconstructed with the selected parameters.

The analysis of the flow statistics reported in Fig. 8 enables estimates of flow parameters needed as input for the LiSBOA. For instance, the wake region is characterized by $\sqrt{\langle \overline{u'^2} \rangle}/U_\infty \approx 0.1$ and $\tau U_\infty/D \approx 0.4$ ($\tau \approx 5$ s).

A main limitation of LiDARs is represented by the spatio-temporal averaging of the velocity field, which is connected with the acquisition process. Three different types of smoothing mechanisms can occur during the LiDAR sampling: the first is the averaging along the laser beam direction within each range gate, which has been commonly modeled through the convolution of the actual velocity field with a weighting function within the measurement volume (Smalikho, 1995; Frehlich, 1997; Sathe et al., 2011). The second process is the time-averaging associated with the sampling period required to achieve a back-scattered signal with adequate intensity (O'Connor et al., 2010; Sathe et al., 2011), while the last one is the transverse averaging (azimuth-wise or elevation-wise) occurring in case of a scanning LiDAR operating in continuous mode (Stawiarski et al., 2013). These filtering processes lead to significant underestimation of the turbulence intensity (Sathe et al., 2011), overestimation of integral length-scales (Stawiarski et al., 2015), and damping of energy spectra for increasing wavenumbers (Risan et al., 2018; Puccioni and Iungo, 2020).

Three versions of a LiDAR simulator are implemented for this work: the simplest one is referred to as ideal LiDAR, which samples the LES velocity field at the experimental points through a nearest-neighbor interpolation. This method minimizes the turbulence damping while retaining the geometry of the scan and the projection of the wind velocity vector onto the laser beam direction. The second version of the LiDAR simulator reproduces a step-stare LiDAR, namely the LiDAR scans for the entire duration of the accumulation time at a fixed direction of the LiDAR laser beam. Two filtering processes take place for this configuration: beam-wise convolution and time averaging. To model the beam-wise average, the retrieval process of the Doppler LiDAR is reproduced using a spatial convolution (Mann et al., 2010):

$$u_{\text{LOS}}(\boldsymbol{x}, t) = \int\limits_{-\infty}^{\infty} \phi(s) \boldsymbol{n} \cdot \boldsymbol{u}(\boldsymbol{x} + \boldsymbol{n}s, t) ds, \tag{15}$$

where $\boldsymbol{n}$ is the LiDAR laser-beam direction, $\boldsymbol{u}$ is the instantaneous velocity vector and the dot indicates scalar product. A triangular weighting function $\phi(s)$ was proposed by Mann et al. (2010):

$$\phi(s) = \begin{cases} \frac{\Delta r/2 - |s|}{\Delta r^2/4} & \text{if } |s| < \Delta r/2 \\ 0 & \text{otherwise}, \end{cases} \tag{16}$$

where $\Delta r$ is the gate length. The former expression is valid assuming matching time windowing, i.e. gate length equal to the pulse width, and the velocity value is retrieved based on the first momentum of the back-scattering spectrum. Despite its

simplicity, Eq. (16) has shown to estimate realistic turbulence attenuation due to the beam-wise averaging process of a pulsed Doppler wind LiDAR (Mann et al., 2009). Furthermore, time-averaging occurs due to the accumulation time necessary for the LiDAR to acquire a velocity signal with sufficient intensity and, thus, signal-to-noise-ratio. This process is modeled through a
window average within the acquisition interval of each beam. For the sampling of the LES velocity field in space and time, a nearest-neighbor interpolation method is used.

The third version of the LiDAR simulator mimics a pulsed LiDAR operating in continuous mode and performing PPI scans, where, in addition to the beam-wise convolution and time-averaging, azimuth-wise averaging occurs due to the variation of the LiDAR azimuth angle of the scanning head during the scan. The latter is taken into account by adding to the time average an
azimuthal averaging among all data points included within the following angular sector:

$$
\begin{cases}
|\theta - \theta_p| < \Delta\theta/2 \\
|\beta - \beta_p| < \sin^{-1}\left(\frac{\Delta z}{2r_p}\right)
\end{cases}
\tag{17}
$$

where $r$ is the radial distance from the emitter, while $\theta$ and $\beta$ are the associated azimuth and elevation angles, respectively. The subscript $p$ refers to the $p$-th LiDAR data point. Following the suggestions by Stawiarski et al. (2013), the out-of-plane thickness, $\Delta z$, is considered equal to the length of the diagonal of a cell of the computational grid.

It is noteworthy that the accuracy estimated through the present analysis only includes error due to the sampling in time and space, and data retrieval. Other error sources, such as the accuracy of the instrument (Rye and Hardesty, 1993; O'Connor et al., 2010), are not included and should be coupled to the LiSBOA estimates for a more general error quantification (Wheeler and Ganji, 2010b).

Figure 10a shows a snapshot of the streamwise velocity field over the horizontal plane at hub height obtained from the LES.
The respective data of the radial velocity obtained from the three versions of the LiDAR simulator by considering a scanning pulsed wind LiDAR deployed at the turbine location and at hub height highlight the increased spatial smoothing of the radial velocity field by adding the various averaging processes connected with the LiDAR measuring process, namely beam-wise, temporal and azimuthal averaging (Fig. 10).

The application of the LiSBOA requires to provide technical specifications of the LiDAR, specifically accumulation time,
$\tau_a$, number of gates, $N_r$, and gate length, $\Delta r$. For this work, these parameters are selected based on the typical settings of LiDARs Windcube 200S and StreamLine XR (El-Asha et al., 2017; Zhan et al., 2019, 2020), namely $\tau_a = 0.5$ s, $N_r = 39$ and $\Delta r = 25$ m. Furthermore, to probe the wake region, a volumetric scan including several PPI scans with azimuth and elevation angles uniformly spanning the range $\pm 10°$ with a constant angular resolution in both azimuth and elevation is selected, while the virtual LiDAR is placed at the turbine hub.

With the information provided about the flow under investigation and the LiDAR system, it is possible to draw the Pareto front for the optimization of the LiDAR scan as a function of different combinations of angular resolutions of the LiDAR scanning head, $\Delta\theta$ and $\Delta\beta$, and the smoothing parameter of the LiSBOA, $\sigma$, as shown in Fig. 11 for the case under investigation.

For the optimization of the LiDAR scan, the LiDAR angular resolution, $\Delta\theta$, is evenly varied, for a total number of 7 cases, from $0.75°$ to $4°$, whereas three values of the ratio $\Delta\beta/\Delta\theta$, namely 0.5, 1 and 2, are tested separately. The four values of $\sigma$

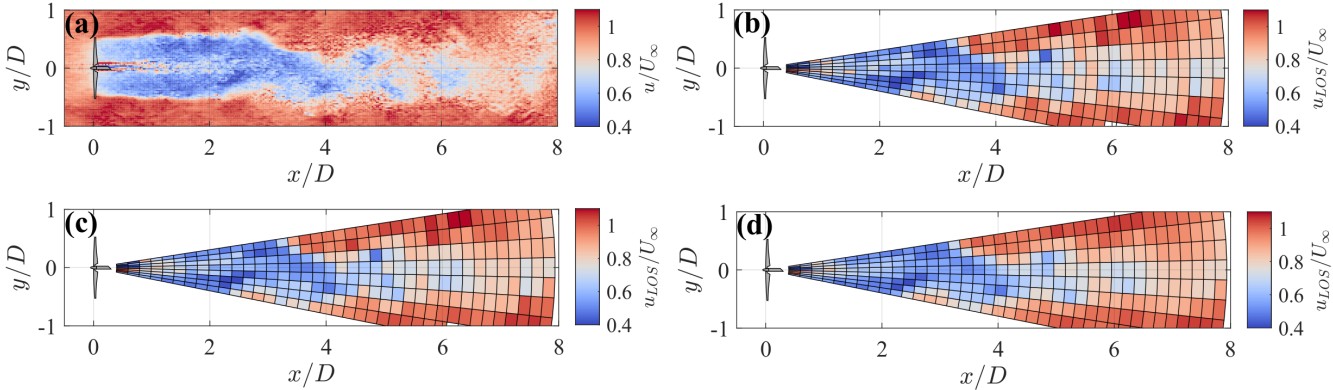

**Figure 10.** Snapshot at the hub-height horizontal plane of the wake generated by the 5-MW NREL reference wind turbine: **(a)** LES streamwise velocity; **(b)** ideal virtual LiDAR with angular resolution $\Delta\theta = 2.5°$, zero elevation, accumulation time $\tau_a = 0.5$ s, gate length $\Delta r = 25$ m; **(c)** step-stare virtual LiDAR (same settings); **(d)** continuous mode virtual LiDAR (same settings).

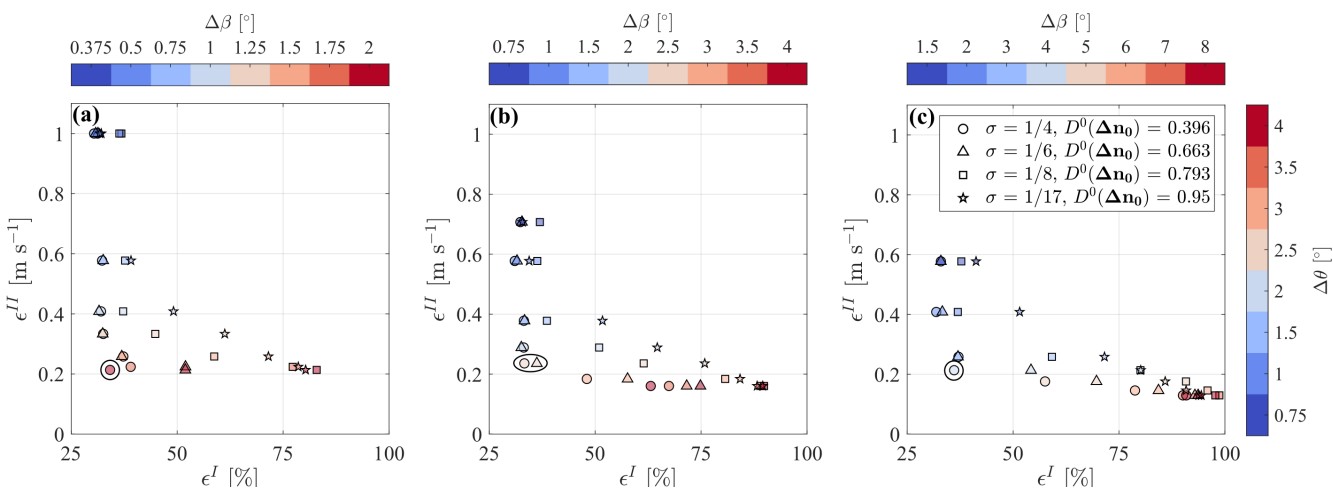

**Figure 11.** Pareto front for the design of the optimal LiDAR scan for the LES dataset for different $\Delta\theta/\Delta\beta$ combinations: **(a)** $\Delta\beta/\Delta\theta = 0.5$; **(b)** $\Delta\beta/\Delta\theta = 1$; **(c)** $\Delta\beta/\Delta\theta = 2$. The circle indicates the selected optimal configurations.

recommended in Table 2 to achieve a response of the mean $D^m(\boldsymbol{\Delta\tilde{n}_0}) = 0.95$ are considered here. In Fig. 11, markers indicate the different $\sigma$ and, thus, the response of high-order statistical moments, $D^0(\boldsymbol{\Delta\tilde{n}_0})$. Changing the ratio $\Delta\beta/\Delta\theta$ affects the optimal $\Delta\theta$ (circled in black in Fig. 11); however, it has a negligible effect on the magnitude of the optimal $\epsilon^I$ and $\epsilon^{II}$. For the rest of the discussion, we select the setup $\Delta\beta/\Delta\theta = 1$, as suggested by Fuertes Carbajo and Porté-Agel (2018). The Pareto front for $\Delta\beta/\Delta\theta = 1$ (Fig. 11b) shows that increasing $\Delta\theta$ from $0.75°$ up to $2.5°$ drastically reduces the uncertainty on the mean ($\epsilon^{II}$)

by roughly 70 %, but does not affect significantly data loss consequent to the enforcement of the Petersen-Middleton constraint ($\epsilon^I$). For larger angular resolutions, the statistical significance improves just marginally, but at the cost of a relevant data loss.

For $\Delta\theta \geq 2°$, in particular, $\epsilon^I$ becomes extremely sensitive to $\sigma$, with the most severe data loss occurring for small $\sigma$ (i.e. small $R_{\max}$). The Pareto front also shows that to achieve a higher response for the higher-order statistics, $D^0(\Delta\tilde{\boldsymbol{n}}_0)$, generally entails an increased data loss and/or statistical uncertainty of the mean. This analysis suggests that the optimal LiDAR scan for the reconstruction of the mean velocity field should be performed with $\Delta\theta = 2.5°$ and $\sigma = 1/4$ or $1/6$.

Virtual LiDAR simulations are performed for all the values of angular resolution utilized in the Pareto front reported in Fig. 11. The streamwise component is estimated from the line-of-sight velocity through an equivalent velocity approach (Zhan et al., 2019). The latter states that, for small elevation angles (i.e. $\beta \ll 1$) and under the assumptions of negligible vertical velocity compared to the horizontal component (i.e. $|w| \ll \sqrt{u^2 + v^2}$) and uniform wind direction, $\theta_w$, a proxy for the streamwise velocity can be calculated as:

$$u \sim \frac{u_{\text{LOS}}}{cos(\theta - \theta_w)cos\beta}. \tag{18}$$

The mean velocity and turbulence intensity are reconstructed through the LiSBOA. The maximum error is quantified through the 95-th percentile of the absolute error, $AE_{95}$, using as reference the LES statistics interpolated on the LiSBOA grid.

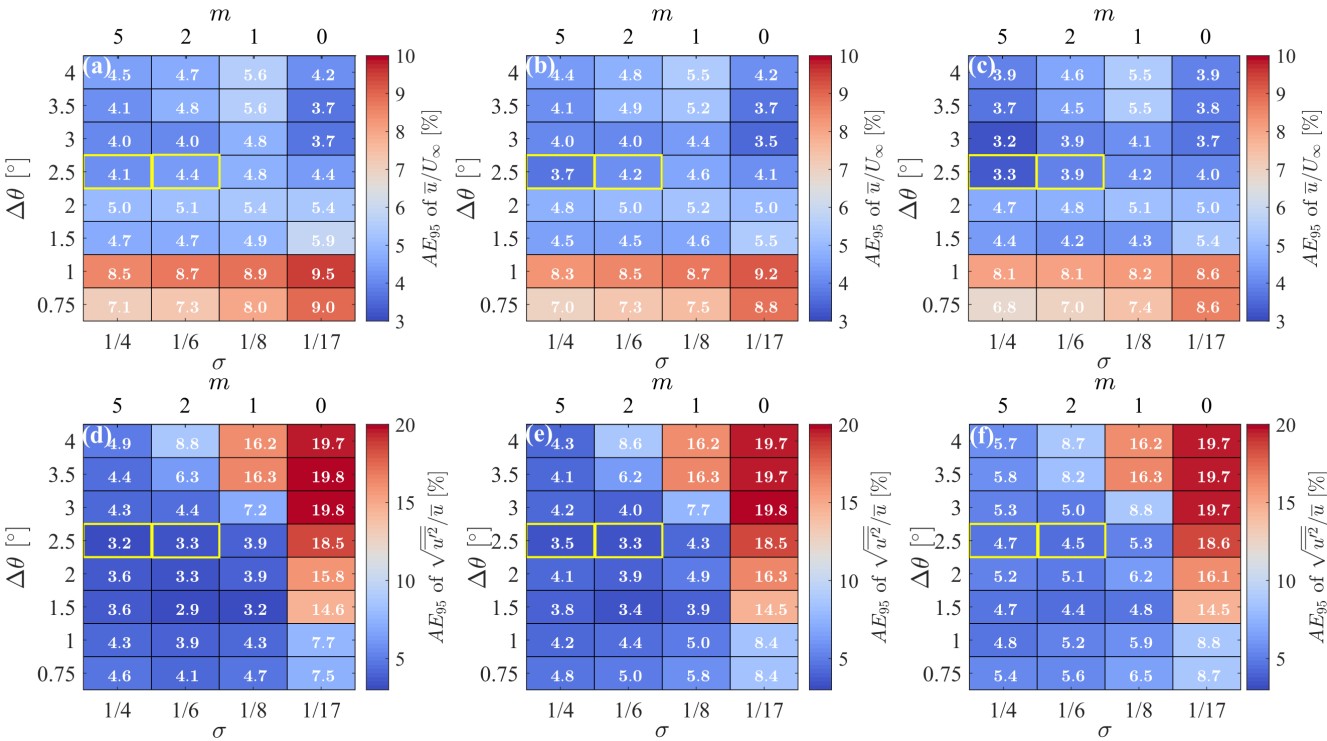

**Figure 12.** Error analysis of the LiSBOA applied to virtual radial velocity fields: **(a)** and **(d)** ideal LiDAR; **(b)** and **(e)** step-stare LiDAR; **(c)** and **(f)** continuous LiDAR; **(a)**, **(b)** and **(c)** mean streamwise velocity; **(d)**, **(e)** and **(f)** streamwise turbulence intensity. The optimal configurations are highlighted in yellow.

Fig. 12 reports the $AE_{95}$ for the flow statistics for all the virtual experiments. The error for the mean field (Figs. 12a-c) is mostly governed by the angular resolution, with a higher error occurring for slower scans. This is a clear consequence of the increased statistical uncertainty due to the limited number of scan repetitions, $L$, that are achievable for small $\Delta\theta$ values and a fixed total sampling period, $T$, while $AE_{95}$ stabilizes for $\Delta\theta \geq 2.5°$. The trend of the $AE_{95}$ for $\overline{u}/U_\infty$ with the pair smoothing parameter-number of iterations, $\sigma - m$, is less significant since the theoretical response of the fundamental mode is ideally equal for all the four cases. Conversely, the error on the turbulence intensity (Figs. 12d-f) shows low sensitivity to the angular resolution but a steep increase for small $\sigma$ values, which is due to the reduction of the radius of influence, $R_{\text{max}}$, and the number of points averaged per grid node.

From a more technical standpoint, the error on the mean velocity field, $\overline{u}/U_\infty$, appears to be relatively insensitive to the type of LiDAR scan, with the spatial and temporal filtering operated by the step-stare and continuous LiDAR being even beneficial in some cases. In contrast, the error on the turbulence intensity exhibits a more consistent and opposite trend, with the continuous LiDAR showing the most severe turbulence damping. This feature has been extensively documented in previous studies, see e.g. Sathe et al. (2011).

This error analysis confirms that the optimal configurations selected through the Pareto front (i.e. $\Delta\theta = 2.5°$, $\sigma = 1/4 - m = 5$ and $\sigma = 1/6 - m = 2$) are arguably optimal in terms of accuracy ($AE_{95}$ of $\overline{u}/U_\infty = 3.3 - 4.1\%$ and $3.9 - 4.4\%$, $AE_{95}$ of $\sqrt{\overline{u'^2}}/\overline{u} = 3.2 - 4.7\%$ and $3.3 - 4.5\%$, respectively) and data loss ($\epsilon^I = 33\%$ and $37\%$, respectively).

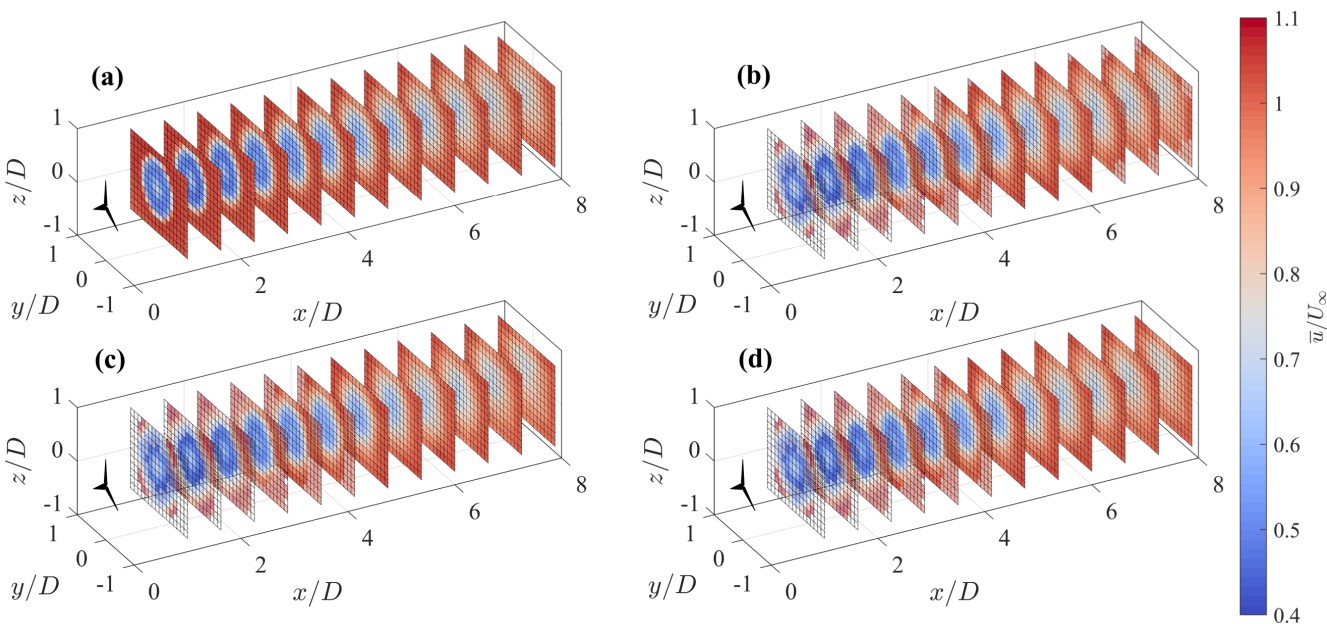

**Figure 13.** Mean streamwise velocity for $\Delta\theta = 2.5°$, $\sigma = 1/4$, $m = 5$: **(a)** LES; **(b)** ideal; **(c)** step-stare; **(d)** continuous mode LiDAR. The shaded area corresponds to the points rejected after the application of the Petersen-Middleton constraint.

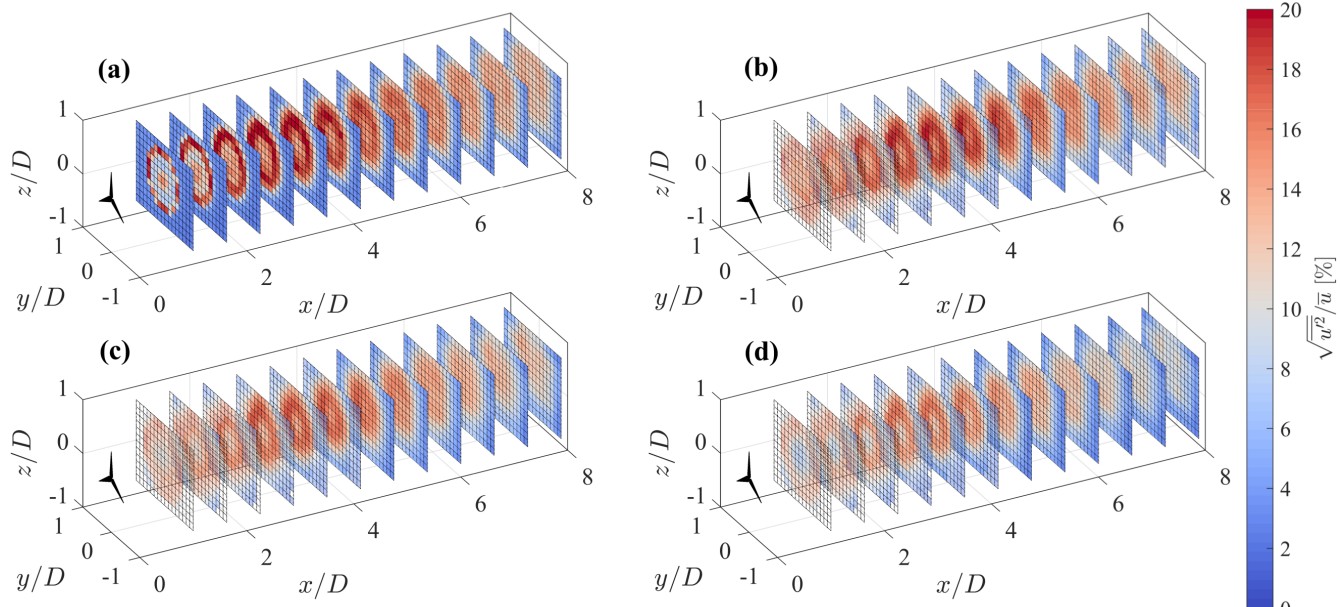

**Figure 14.** As in Fig. 13 but for streamwise turbulence intensity.

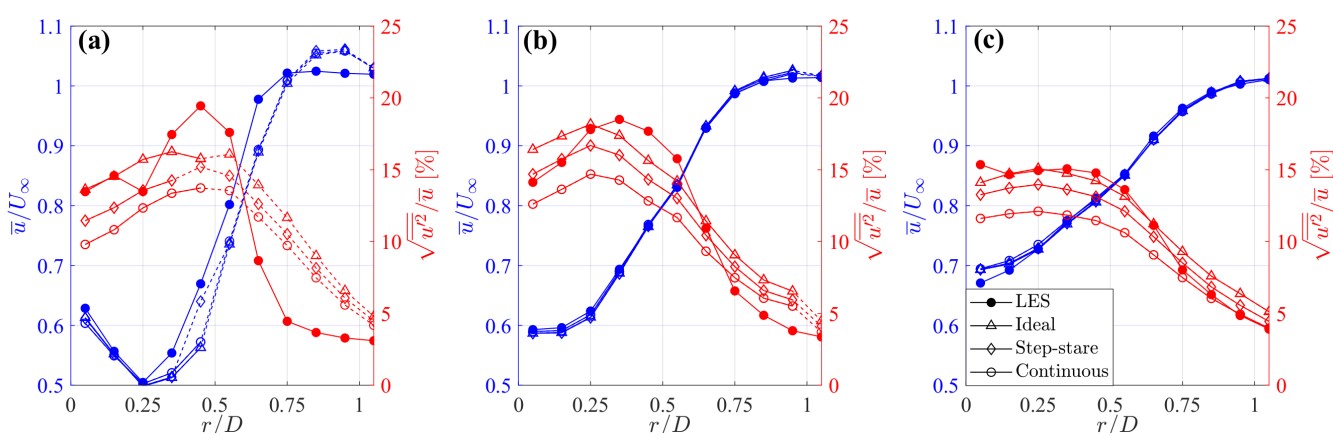

**Figure 15.** Azimuthally averaged profiles of mean streamwise velocity and turbulence intensity for three downstream locations: **(a)** $x/D = 2.25$; **(b)** $x/D = 4.125$; **(c)** $x/D = 6$. The dashed lines correspond to regions rejected after the application of the Petersen-Middleton constraint.

The 3D fields of mean velocity and turbulence intensity calculated over $T = 750$ s through the first optimal configuration, (i.e. $\Delta\theta = 2.5°$, $\sigma = 1/4$, $m = 5$), are rendered in Figs. 13 and 14, respectively. Furthermore, in Fig. 15, azimuthally-averaged profiles at three downstream locations are also provided for a more insightful comparison. The mean velocity field is reconstructed fairly well regardless of the type of LiDAR scan, due to the careful choice of the fundamental half-wavelength, $\boldsymbol{\Delta n_0}$,

for this specific flow. On the other hand, the reconstructed turbulence intensity is highly affected by the LiDAR processing, which leads to visible damping of the velocity variance for the step-stare and even more for the continuous mode. The ideal LiDAR scan, whose acquisition is inherently devoid of any space-time averaging, allows retrieving the correct level of turbulence intensity for locations for $x \geq 4D$, while in the near wake it struggles to recover the thin turbulent ring observed in the wake shear layer. Indeed, such a short-wavelength feature has a small response for the chosen settings of the LiSBOA, in particular $\Delta n_0$ and $\sigma$ (see Fig. 3b). On the other hand, any attempt to increase the response of the higher-order moments, for instance by reducing the fundamental half-wavelengths or decreasing the smoothing and the number of iterations, would result in higher data loss and fewer experimental points per grid node.

Finally, Figs. 16 and 17 show $\overline{u}/U_\infty$ and $\sqrt{\overline{u'^2}}/\overline{u}$ over several cross-flow planes and for all the combinations of $\sigma - m$ tested for the ideal LiDAR and the optimal angular resolution. For the mean velocity, the most noticeable effect is the increasingly severe data loss consequent to the reduction of $\sigma$, which indicates $\sigma = 1/4$ - $m = 5$ as the most effective setting. The turbulence intensity exhibits, in addition to the data loss, a moderate increase in the maximum value for smaller $\sigma$, which is due to the higher response of the higher-order statistics (see Table 2). However, this effect is negligible in the far wake were the radial diffusion of the initially sharp turbulent shear layer results in a shift of the energy content towards scales with larger $\Delta n$, which are fairly recovered even for $\sigma = 1/4$.

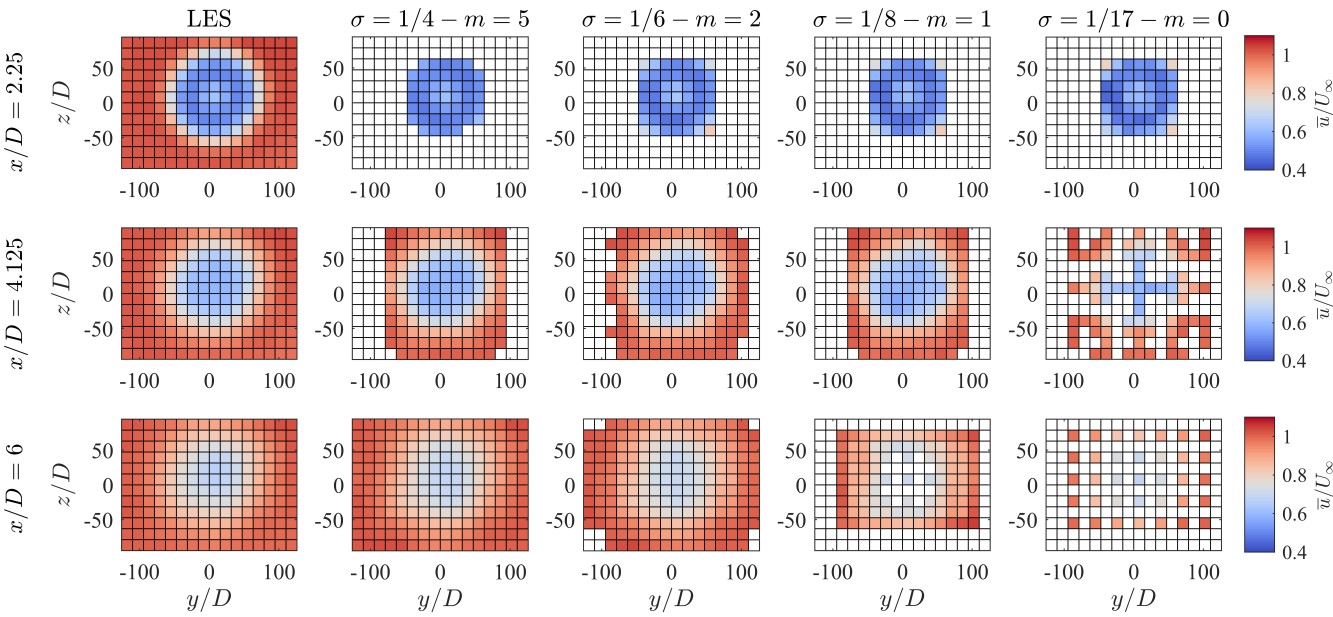

**Figure 16.** Mean streamwise velocity fields obtained through the ideal LiDAR simulator with $\Delta \theta = 2.5°$ over cross-flow planes at three downstream locations and four combinations of $\sigma - m$, compared with the corresponding LES data.

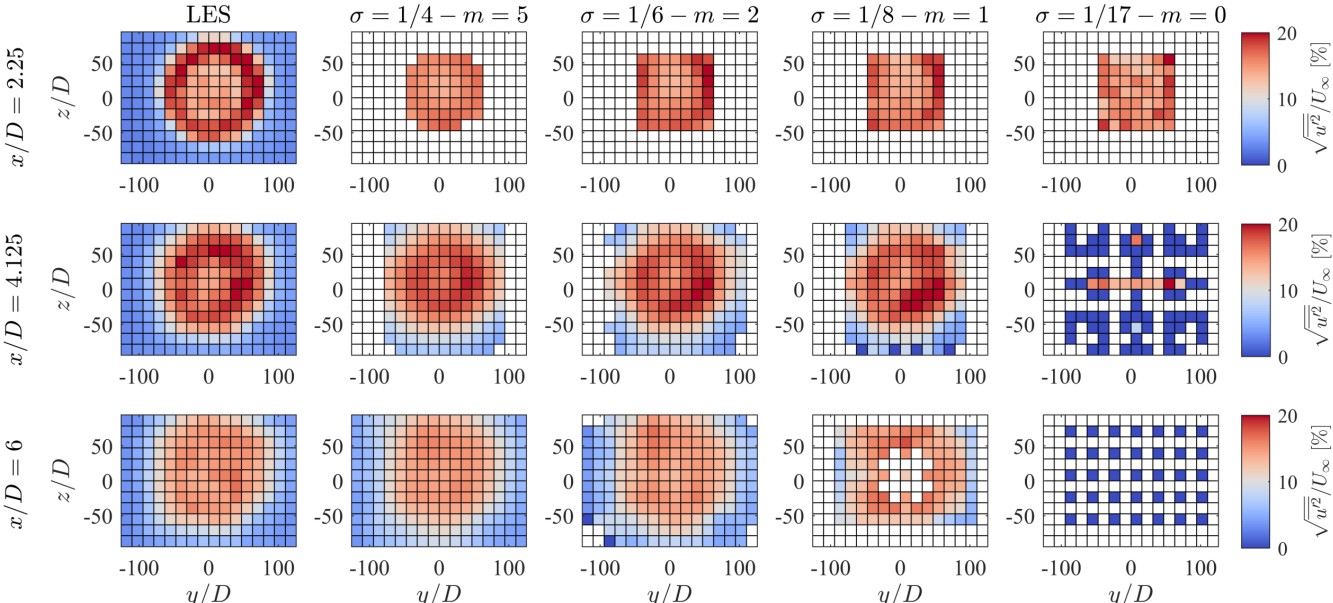

**Figure 17.** Same as Fig. 16 but for streamwise turbulence intensity.

## 6 Notes on LiSBOA applications

The LiSBOA can be applied to LiDAR datasets that are statistically homogeneous as a function time, $t$. This statistical property can be ensured with two approaches. The first approach consists in considering LiDAR data collected continuously in time with a given sampling frequency for a period where environmental parameters, such as wind speed and direction, Obukhov length and Bulk Richardson number for the atmospheric stability regime, are constrained within prefixed intervals (e.g. Banta et al. (2006); Iungo et al. (2013b); Kumer et al. (2015); Puccioni and Iungo (2020)). For instance, the statistical stationarity of a generic flow signal, $\alpha$, can be verified through the non-stationary index (IST) (Liu et al., 2017):

$$IST = \frac{|\overline{\alpha'\alpha'}_m - \overline{\alpha'\alpha'}|}{\overline{\alpha'\alpha'}}, \tag{19}$$

where $\overline{\phantom{x}}$ represents time-averaging and $\overline{\alpha'\alpha'}_m$ is the mean value of the variance calculated over consecutive non-overlapping sub-periods. The $IST$ values should be lower than a selected threshold depending on the specific flow parameter considered (Foken et al., 2004). A second approach to ensure statistical homogeneity of the LiDAR dataset consists in performing a cluster analysis based on environmental parameters, such as those mentioned above. This can be a fruitful alternative when the application of the first approach leads to too short periods with statistical stationarity and, thus, with low accuracy in the calculation of the turbulent statistics. With the clustering approach, larger datasets can be achieved for each cluster enabling an enhanced statistical convergence (see e.g. applications of clustering analysis to LiDAR measurements of wind turbine wakes Machefaux et al. (2016); Bromm et al. (2018); Iungo et al. (2018); Zhan et al. (2019, 2020)).

The results of the LiSBOA for optimal design a wind LiDAR scans are affected by the selection of the input parameters, such as the total sampling time, $T$, the integral time-scale, $\tau$, the velocity variance, $\overline{u'^2}$, the fundamental half-wavelength, $\boldsymbol{\Delta n_0}$. In this section, we will discuss the sensitivity of the LiSBOA to these input parameters by considering as a reference case the volumetric scan performed with the virtual LiDAR technique on the LES dataset analyzed in Sect. 5. The respective results are summarized in Fig. 18.

The total sampling time, $T$, directly affects the objective function $\epsilon^{II}$ through parameter $L$, which represents the number of realizations. In Fig. 18a-d, different Pareto fronts are generated for the case under investigation by varying $T$ from 3 minutes to 1 hour. The various Pareto fronts exhibit similar trends for the various values of $T$, and generally higher values of $\epsilon^{II}$, so lower statistical accuracy, for smaller $T$. For all the cases, the optimal configuration is still that selected in Sect. 5, namely $\Delta\theta = 2.5° - \sigma = 1/4$.

For this sensitivity study, the characteristic integral time-scale, $\tau$, has been varied between 0 s (completely random uncorrelated data) up to 35 s, the upper value being based on the largest integral length scale in the ABL according to (ESDU, 1975) and considering an advection velocity of 8 m s$^{-1}$. The respective Pareto fronts reported in Fig. 18e-h show that the optimal LiDAR scan is weakly affected by variations of $\tau$, which is an advantageous feature of the LiSBOA for applications where this parameter cannot be estimated from previous investigations or literature.

Regarding the characteristic velocity variance, $\overline{u'^2}$, it is a multiplicative parameter for the objective function $\epsilon^{II}$ (Eq. (14)). Therefore, even though it affects the accuracy of the statistics retrieved, it does not alter the selection of the optimal scanning parameters.

The choice of the fundamental half-wavelength, $\boldsymbol{\Delta n_0}$, deserves special attention since it affects both the optimal scan design and retrieval of data statistics. The fundamental half-wavelength can be considered as the cut-off wavelength of the spatial low-
pass filtering operated by the LiSBOA (Sect. 4). The selection of $\boldsymbol{\Delta n_0}$ depends mostly on the length of the smallest spatial feature of interest in the flow under investigation, so the Pareto front is likely to be rather sensitive to changes in $\boldsymbol{\Delta n_0}$. As mentioned in Sect. 4, if the fundamental half-wavelengths are too large compared to the predominant spatial modes, the turbulence statistics may be contaminated by over-smoothing and dispersive stresses, whereas overly small $\boldsymbol{\Delta n_0}$ may require too small angular and radial resolutions, a longer sampling period, and, thus, a smaller number of repetitions for a given $T$. Fig.
18i-l shows the Pareto fronts calculated at different $\Delta n_{0,x}$. The previously selected optimal setup ($\Delta\theta = 2.5° - \sigma = 1/4$) still belongs to the optimality frontier. Finally, Fig. 18p-m displays the effect of varying $\Delta n_{0,y}$ on the Pareto front. Unlike the other cases, the shape of the front is very sensitive to $\Delta n_{0,y}$, with a significant increase of data loss, $\epsilon^I$, consequent to refinements of the angular resolution, $\Delta\theta$. The Pareto front correctly indicates that finer angular resolutions are needed to adequately sample a velocity field characterized by smaller wavelengths.

For the sake of completeness, the influence of the different fundamental half-wavelengths on the statistics is assessed by calculating the $AE_{95}$ between LiSBOA and LES for the statistics reconstructed using several combinations of $\Delta n_{0,x}$ and $\Delta n_{0,y}$ (Fig. 19). Larger values of $\Delta n_{0,x}$ and $\Delta n_{0,y}$ produce detrimental effects on the accuracy for both mean velocity and turbulence intensity due to the over-smoothing of the mean velocity and turbulence intensity field, and dispersive stresses, for the turbulence intensity only. On the other hand, excessively small values of $\Delta n_{0,y}$ exhibit a slight increase in error as a

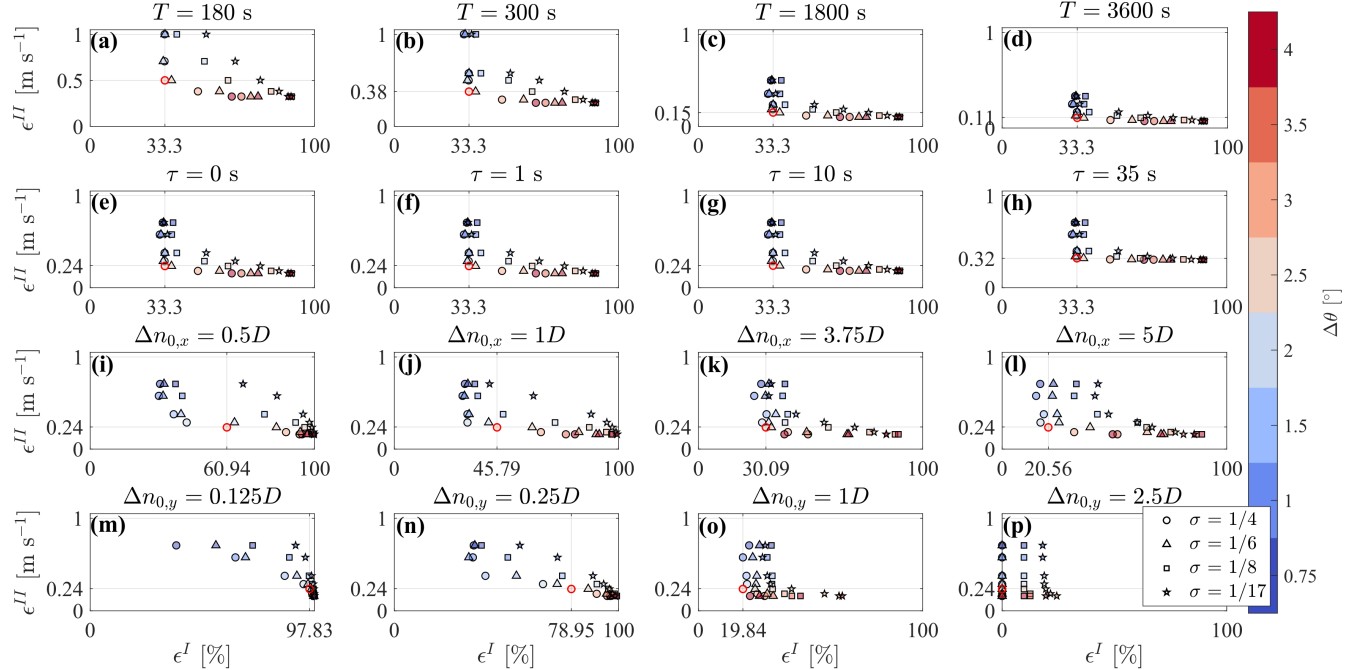

**Figure 18.** Pareto fronts for the design of the volumetric scan for different inputs: **(a-d)**, sensitivity to total sampling time $T$; **(e-h)**, sensitivity to integral time-scale, $\tau$; **(i-l)**: sensitivity to streamwise fundamental half-wavelength, $\Delta n_{0,x}$; **(m-p)**: sensitivity to spanwise fundamental half-wavelength, $\Delta n_{0,y}$. The parameters not indicated on the top of the figures are kept equal to the optimal design case identified in Sect. 5.

consequence of the smaller number of samples per grid node. Nonetheless, the most relevant effect, in this case, is represented by the high data loss, as already identified in the Pareto front (e.g. Fig. 18m and n). It is noteworthy how the choice of $\Delta n_0 = [2.5, 0.5, 0.5]D$ done in Sect. 5, which was purely based on physical considerations about the expected relevant modes in the near wake, turned out to be the optimal configuration in terms of the overall error of $\overline{u}/U_\infty$ and $\sqrt{\overline{u'^2}}/\overline{u}$.

We acknowledge that the technical specifications required by LiSBOA (namely $\tau_a$, $N_r$, $\Delta r$) are dependent on the specific
LiDAR system used, the contingent atmospheric conditions and the best practices followed by the user. Since these parameters are greatly case-dependent, they will not be discussed further in this context. In general, the selection of the accumulation time and gate length is a trade-off between the need to achieve a target maximum range, while keeping a sufficiently fine radial resolution and sufficient intensity of the back-scattered LiDAR signal. In case of uncertain environmental conditions, it is recommended to check before the deployment the influence of selected combinations of $\tau_a$, $N_r$, $\Delta r$ on the Pareto front.

For the sake of completeness, LiSBOA is compared with other widely-used techniques for statistical post-processing of wind LiDAR data, specifically Delaunay triangulation (see e.g. Clive et al. (2011); Trujillo et al. (2011); Iungo and Porté-Agel (2014); Machefaux et al. (2015); Trujillo et al. (2016)), linear interpolation in spherical coordinates (see e.g. Mohr and Vaughan (1979); Fuertes Carbajo and Porté-Agel (2018)), and window averaging (see e.g. Newsom et al. (2008)). Figure 20 shows the mean velocity and turbulence intensity fields retrieved from the considered LES dataset through the various techniques for

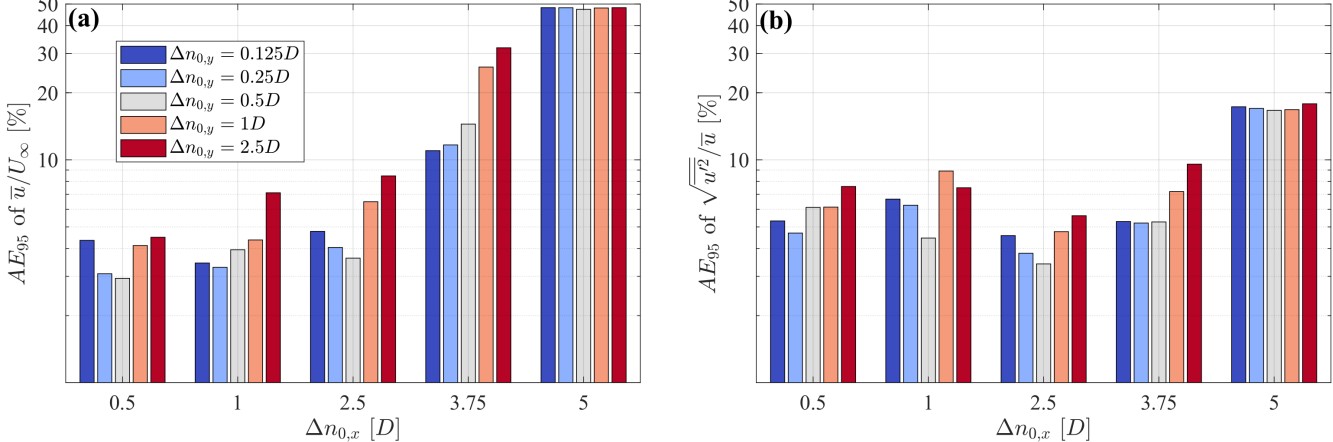

**Figure 19.** $AE_{95}$ of the statistics reconstructed from virtual LiDAR data for different streamwise and spanwise fundamental half-wavelengths, for the setup $\Delta\theta = 2.5° - \sigma = 1/4$: **(a)** mean streamwise velocity; **b** streamwise turbulence intensity.

a step-stare virtual-LiDAR scan. The various methods use the same grid as for the LiSBOA (see Sect. 5). The voids in the 3D rendering correspond to regions outside of the data distribution for Delaunay triangulation and linear interpolation (i.e. extrapolation cannot be performed), or bins having a standard error on the mean higher than 15% of the incoming wind speed for the window average (in analogy with LiSBOA, see Fig. 4). A qualitative comparison of the results reported in Fig. 20 with those for the LiSBOA in Figs. 13(c) and 14(c) reveals that LiSBOA is the method enabling the largest spatial coverage for the

retrieved statistics for the same LiDAR scan. From the parameter $\epsilon_I$, which is reported in Table 3 for the various methods, it is noted that the LiSBOA is the method with the lowest data rejection rate ($\epsilon_I$=33%), while the largest is for the window average ($\epsilon_I$=66%).

     Overall all the methods, except the window average, have similar accuracy in the retrieval of the mean velocity (see the absolute percentage error, MAPE, in Table 3), yet LiSBOA is the method with the lowest error. Furthermore, LiSBOA is the

only method not showing artifacts for the retrieval of turbulence intensity over space, such as enhanced turbulence intensity and unexpected peaks, as the significantly lower error on turbulence intensity confirms. This result is in agreement with Trapp and Doswell (2000), where the effectiveness of the Barnes scheme in the suppression of short-wavelength noise compared to linear interpolation was already noted. Finally, the computational time using Matlab on a quad-core i7 laptop is negligible and comparable for all the algorithms considered ($\sim$ 1-2s), with only the linear interpolation being considerably faster (Table 3).

On a final note, it is noteworthy that the LiSBOA is currently formulated for a single scalar field, namely a velocity component (radial or equivalent horizontal). However, in principle, this procedure can be extended to vector fields, such as fully 3D velocity fields. Furthermore, other constraints can be added for the optimal scanning design, such as imposing a divergence-free velocity field for incompressible flows. Also, some modifications could extend the application of the LiSBOA to other remote sensing instruments, such as sodars and scanning radars.

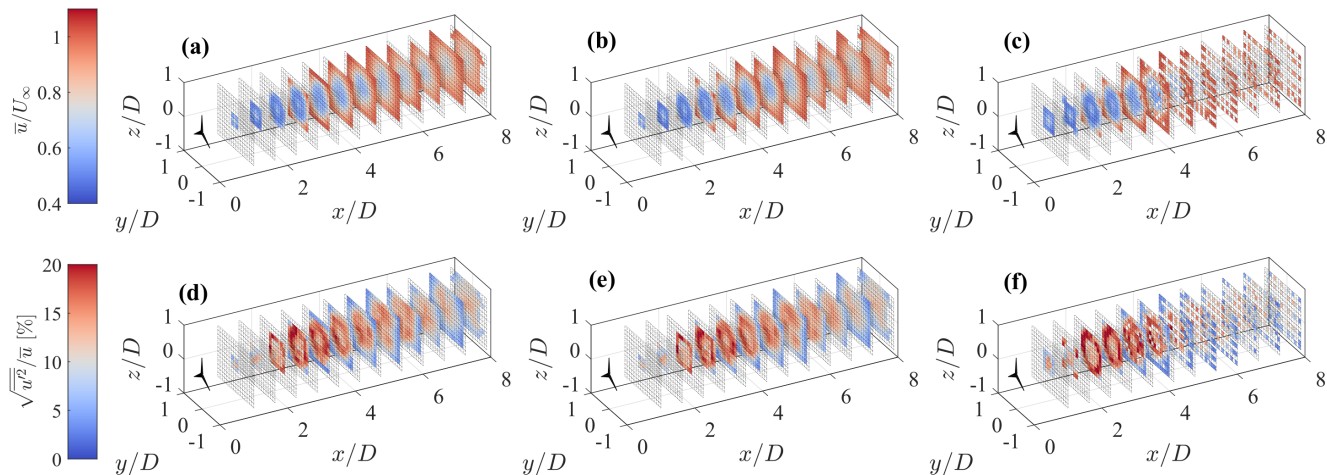

**Figure 20.** Statistics retrieved from a step-stare virtual LiDAR scan of the LES dataset by means of different techniques: **(a)**, **(d)** Delaunay triangulation; **(b)**, **(e)**, linear interpolation; **(c)**, **(f)** window averaging; **(a)**, **(b)**, **(c)**, mean streamwise velocity; **(d)**, **(e)**, **(f)** streamwise turbulence intensity.

**Table 3.** Comparison between LiSBOA and other techniques for the retrieval of mean velocity and turbulence intensity from the LES dataset through a virtual LiDAR scan.

|  | $MAPE$ of $\overline{u}/U_\infty$ | $MAPE$ of $\sqrt{\overline{u'^2}}/\overline{u}$ | $\epsilon^I$ | Time |
|---|---|---|---|---|
| LiSBOA | 1.2% | 1.3% | 33% | 1.79 s |
| Delaunay triangulation | 1.3% | 1.7% | 53% | 0.65 s |
| Linear interpolation | 1.3% | 1.6% | 55% | 0.01 s |
| Window average | 2% | 1.5% | 66% | 1.55 s |

## 7 Conclusions

A revisited Barnes objective analysis for sparse, non-uniformly distributed and stationary LiDAR data has been formulated to calculate mean, variance, and higher-order statistics of the wind velocity field over a structured N-dimensional Cartesian grid. This LiDAR Statistical Barnes Objective Analysis (LiSBOA) provides a theoretical framework to quantify the response in the reconstruction of the velocity statistics as a function of the spatial wavelengths of the velocity field under investigation and quantification of the sampling error.

The LiSBOA has been validated against volumetric synthetic 3D data generated through Monte-Carlo simulations. The results of this test have shown that the sampling error for a monochromatic scalar field is mainly driven by the data spacing normalized by the half-wavelength.

The LiSBOA framework provides guidelines for the optimal design of scans performed with a scanning Doppler pulsed wind LiDAR and calculation of wind velocity statistics. The optimization problem consists in providing background information about the turbulent flow under investigation, such as total sampling time, expected velocity variance and integral length scales, technical specifications of the LiDAR, such as range gate and accumulation time, and spatial wavelengths of interest for the velocity field. The formulated optimization problem has two cost functions, namely the percentage of grid nodes not satisfying the Petersen-Middleton constraint for the smallest half-wavelength of interest (i.e. lacking adequate spatial resolution to avoid aliasing in the statistics), and the standard deviation of the sample mean. The output of the optimization problem are the LiDAR angular resolution and, for a given response of the mean field, the allowable smoothing parameters and number of iterations to use for the LiSBOA.

The LiSBOA has been validated using a dataset obtained through the virtual LiDAR technique, namely by numerically sampling the turbulent velocity field behind the rotor of a 5 MW turbine obtained from a large eddy simulation (LES). The 3D mean streamwise velocity and turbulence intensity fields have shown a maximum error with respect to the LES dataset of about $4\%$ of the undisturbed wind speed for the mean streamwise velocity and $4\%$ of the streamwise turbulence intensity, in absolute terms. Wake features, such as the high-velocity stream in the hub region and the turbulent shear layer at the wake boundary, have been accurately reconstructed.

In the companion paper (Letizia et al., 2020), the LiSBOA is used to reconstruct the turbulence statistics of utility-scale turbine wakes probed with scanning pulsed Doppler LiDARs. That study also illustrates the detailed preconditioning applied to the raw LiDAR data to extract statistically stationary and normalized radial velocity data and showcases the potential of LiSBOA for wind energy research.

*Code availability.* The LiSBOA algorithm is implemented in a publicly available code which can be downloaded at the following URL: https://www.utdallas.edu/windflux/software-datasets/.

**Appendix A: Derivation of the analytical response function of the LiSBOA**

The first iteration of the LiSBOA produces a weighted average in space of the scalar field, $f$, with the weights being Gaussian functions centered at the specific grid nodes, $\boldsymbol{x}$. In the limit of a continuous function defined over an infinite domain, Eq. (3) represents the convolution between the scalar field, $f$, and the Gaussian weights, $w$. Therefore, the response function of the LiSBOA, can be expressed in the spectral domain as (Pauley and Wu, 1990):

$$D^0 = \frac{\mathfrak{F}[g^0]}{\mathfrak{F}[f]} = \mathfrak{F}[w], \tag{A1}$$

where the operator $\mathfrak{F}$ indicates the Fourier transform (FT). The FT of the weighting function in Eq. (A1) can be conveniently recast as the product of N one-dimensional FT:

$$\mathfrak{F}[w] = \prod_{p=1}^{N} \int_{-\infty}^{\infty} \frac{1}{\sqrt{2\pi}\sigma} e^{\frac{-x_p^2}{2\sigma^2}} \cdot e^{-\mathrm{i}k_p x_p} dx_p, \tag{A2}$$

where $k_p$ is the directional wavenumber and $\mathrm{i} = \sqrt{-1}$. Hence, by leveraging the closed-form FT of the Gaussian function
(Greenberg, 1998):

$$\mathfrak{F}\left[ \frac{1}{\sqrt{2\pi}\sigma} e^{\frac{-x^2}{2\sigma^2}} \right] = e^{\frac{-k^2\sigma^2}{2}} \tag{A3}$$

we get the desired results, i.e.:

$$D^0(\boldsymbol{k}) = e^{-\frac{\sigma^2}{2}|\mathbf{k}|^2}. \tag{A4}$$

*Author contributions.* SL and GVI developed the LiSBOA and prepared the manuscript. The LiDAR data were generated as a team effort
including contributions from all three authors. SL implemented the LiSBOA in a Matlab code under the supervision of GVI.

*Competing interests.* No competing interests are present.

*Acknowledgements.* This research has been funded by a grant from the National Science Foundation CBET Fluid Dynamics, award number 1705837. This material is based upon work supported by the National Science Foundation under grant IIP-1362022 (Collaborative Research: I/UCRC for Wind Energy, Science, Technology, and Research) and from the WindSTAR I/UCRC Members: Aquanis, Inc., EDP Renewables,
Bachmann Electronic Corp., GE Energy, Huntsman, Hexion, Leeward Asset Management, LLC, Pattern Energy, EPRI, LM Wind, Texas Wind Tower and TPI Composites. Any opinions, findings, and conclusions or recommendations expressed in this material are those of the authors and do not necessarily reflect the views of the sponsors. Texas Advanced Computing Center is acknowledged for providing computational resources. The authors thank Stefano Leonardi and Umberto Ciri for sharing the LES dataset.

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
