# Peer review of "LiSBOA: LiDAR Statistical Barnes Objective Analysis for optimal design of LiDAR scans and retrieval of wind statistics. Part I: Theoretical framework"

_Atmospheric Measurement Techniques, 2020_

## Referee Comment (RC1) · Anonymous Referee #1 · 9 Oct 2020

Within this manuscript the authors present a modified version of a Barnes Objective Analysis technique that is uniquely designed to reconstruct atmospheric properties from remote sensing data. After presenting the technique, the authors discuss how it can be applied to Doppler lidar data to determine the spatial variability of the flow and its features (i.e., variability) by analyzing repeated scanning measurements over a period of time. To this reviewer who is familiar with boundary-layer process and Doppler lidar but has little expertise in use of the Barnes analysis, the technique seems novel and shows promise in enabling new analyses with Doppler wind lidar. This paper will

certainly be a nice contribution to the field by revising an expanding the capabilities of an established technique, and AMT is an appropriate journal for the material. However, there are several significant limitations in using the technique that the authors need to more clearly and directly address. This will likely require significant additional discussion to the paper to address the concerns listed below, prompting major revisions.

Specific Comments

a) Line 1: I think the name LiDAR Statistical Barnes Objective Analysis (LiSBOA) is a little misleading. In reading through the description of the technique, it's unclear to this reader what makes this technique particularly applicable to LiDAR, as it seems like it could be used to a large number of remote sensing measurements. Is this an incorrect assumption? In fact, there is an entire section (Sect. 4) devoted to explaining how this technique can be applied to wind LiDAR data. This technique could actually be of use to researchers using other instruments, as well.

b) Line 22: One primary systematic error of cup anemometers is overspeeding (Busch and Kristensen, 1976), which should be mentioned here.

c) It would be beneficial to add a list of symbols to the start of the article, as is done for the companion paper. Not being overly-familiar with the Barnes analysis technique, the symbol meaning was very difficult to keep track of.

d) Line 46: Fog and low clouds should be added to this list of adverse weather conditions.

e) Line 65: Should be 'horizontal homogeneity of the mean flow', as the turbulent portion may vary over the scanning volume.

f) Line 71-76: These sentences seem out of place, considering the previous paragraph discussed methods to estimate turbulence. They would better fit in there.

g) Line 123: Throughout this discussion, it is clear that the Barnes analysis can only be applied to a scalar field, and not a vector field such as a wind field composed of a 3-D

vector (u, v, and w). The method described here is to be applied to Doppler wind lidar measurements, which is actually the projection of the 3-D wind vector along the beam. Please discuss limitations, assumptions, and other issues that need to be considered when applying this technique, which was developed for scalar quantities, to analyzing any components of a vector wind field. Within this discussion, it must be considered that a Doppler lidar only measures the wind vector reprojected to be along the radial, thus for a scan it is actually sensitive to varying components of the wind based on the azimuth and elevation.

h) Line 354: Based on this figure, it seems like the flow characteristics are critical to design the optimal scan. However, often the flow characteristics are not known and are the objective of the study is to quantify the flow characteristics. In the absence of knowing the flow characteristics, how does one determine the ideal scan and retrieval of statistics especially for flows that have not been studied extensively previously?

i) Line 391: Given the strict limitations of being able to use the technique to reconstruct the flow and its properties, it is a bit of a stretch to state that this is likely to become a standardized method for LiDAR data collection and analysis. While the proposed technique seems highly valuable to investigate the properties of low-altitude atmospheric flows with LiDAR, the chief limitation of needing to assume an unvarying mean flow (wind direction and speed unchanging) over the course of the time period of analysis will significantly limit when this technique can be used, especially given that conditions within the PBL are often highly variable particularly during transitional time periods (around sunrise and sunset, most notably). This significant limitation will limit any use for routine automated analysis, but it will certainly be useful in select case studies. This significant limitation must be more directly discussed within the manuscript, to mitigate the misuse of this technique for analyzing flows when it is not appropriate to use. By using this during time periods when the mean conditions are even slightly changing, there will be a significant overestimate of any variance in the flow and other properties (length scales, etc) will also not be representative of the true properties of the flow.

Editorial Corrections

a) Line 26: Should be volume of 0.01 mˆ3.

b) Line 36: This ("In particular . . . anemometry") is an incomplete sentence.

References

Busch, N.E. and Kristensen, L., 1976. Cup anemometer overspeeding. Journal of Applied Meteorology, 15(12), pp.1328-1332.
* * *

---

## Referee Comment (RC2) · Anonymous Referee #2 · 12 Oct 2020

This manuscript presents a new algorithm to retrieve wind speed and turbulence intensity from Doppler lidar measurements. The algorithm is based on Barnes objective analysis, which was developed to interpolate unevenly spaced observations of a scalar field over a regular grid. Here, this method is extended to 3D-fields, i.e. radial velocity from Doppler lidar. Furthermore, the new algorithm is used to optimise Doppler lidar scan strategy for optimal retrieval.

On the other hand, the algorithm requires substantial preliminary information about the flow in the area of interest as well as stationary conditions during the period when the

scans are conducted, which limits the use of the algorithm. Overall, I find this work within the scope of AMT, but there are several major issues that need to be addressed first.

Specific comments

1. Benefits of the new algorithm are not demonstrated. Synthetic data is used to show that this method can retrieve wind speed and turbulence intensity (defined as u'/U), but these parameters can be retrieved with much simpler methods and less previous information about the flow from Doppler lidar data. How much does the new algorithm improve the retrieval?

2. Can this method be used to retrieve information about turbulence in the inertial sub-range, or is it limited to the outer scale of the velocity spectrum?

3. Section 3. It seems that LiSBOA is validated with a velocity field (Fig. 1a), which I cannot imagine being ever observed in the atmosphere. Furthermore, the sampling appears to be completely random points, when a lidar would always observe radial velocity along line of sight, i.e. denser spacing of (radial) observations near the instrument. Is this assessment valid for atmospheric use of LiSBOA? Please consider using the LES data from Part II for the validation. Moving analysis of LES from Part II to Part I would also shorten Part II, which now has 31 figures.

4. L290: "Firstly, it is crucial to estimate the integral quantities of the flow under investigation required for the application of the LiSBOA, such as extension of the spatial domain of interest, characteristic length-scales, integral time-scale, characteristic temporal standard deviation of the velocity, and expected total sampling time, T, which depends on the typical duration of stationary boundary conditions over the domain." How sensitive is LiSBOA to uncertainty in these parameters? How strict is the requirement of stationary boundary conditions? I'm afraid that these requirements will become a severe limitation for the use of LiSBOA.

[Figure]

5. L327-329: "The angular resolution of the LiDAR scanning head, can be selected to modify the angular spacing between consecutive line-of-sights (i.e. the data spacing)" Do you propose to use the same spacing for both azimuth and elevation angles? Or use only one PPI scan? Please include both angles in the optimization process.

6. It seems that instrumental noise is not considered in the algorithm. What is suitable instrumental uncertainty in radial velocity for use in LiSBOA? This is vital information for scan design as sampling time per profile and/or gate length may need to be increased to cover the full area of interest with strong enough signal.

7. Fig. 7. Thank you for the diagram. Please also provide code for the optimisation process using these inputs.

8.L395 The code is not available where stated. Please include the code as supplementary information so that a copy is permanently archived.

---

## Author Comment (AC1) · 25 Nov 2020

Reply to the comments provided by the Anonymous Referee #1 on the manuscript amt-2020-227 entitled "LiSBOA: LiDAR Statistical Barnes Objective Analysis for optimal design of LiDAR scans and retrieval of wind statistics. Part I: Theoretical framework", by S. Letizia, L. Zhan and G.V. Iungo

The authors sincerely thank the referee for the thorough review and the detailed comments. Our replies are reported in the following. References to pages and lines are based on the revised marked-up manuscript.

**Comments:**

Within this manuscript the authors present a modified version of a Barnes Objective Analysis technique that is uniquely designed to reconstruct atmospheric properties from remote sensing data. After presenting the technique, the authors discuss how it can be applied to Doppler lidar data to determine the spatial variability of the flow and its features (i.e., variability) by analyzing repeated scanning measurements over a period of time. To this reviewer who is familiar with boundary-layer process and Doppler lidar but has little expertise in use of the Barnes analysis, the technique seems novel and shows promise in enabling new analyses with Doppler wind lidar. This paper will certainly be a nice contribution to the field by revising and expanding the capabilities of an established technique, and AMT is an appropriate journal for the material. However, there are several significant limitations in using the technique that the authors need to more clearly and directly address. This will likely require significant additional discussion to the paper to address the concerns listed below, prompting major revisions.

R: We thank the Reviewer for the positive feedback on our research strategy and the results achieved. Additional discussion is now added to the manuscript to clarify the comments arisen by the Reviewer.

**Specific Comments:**

a) Line 1: I think the name LiDAR Statistical Barnes Objective Analysis (LiSBOA) is a little misleading. In reading through the description of the technique, it's unclear to this reader what makes this technique particularly applicable to LiDAR, as it seems like it could be used to a large number of remote sensing measurements. Is this an incorrect assumption? In fact, there is an entire section (Sect. 4) devoted to explaining how this technique can be applied to wind LiDAR data. This technique could actually be of use to researchers using other instruments, as well.

R: The Reviewer is correct. The LiSBOA could be generally applied to wind data collected with different measurement techniques, especially scanning remote-sensing instruments. However, the LiSBOA procedure has been tailored specifically for scanning Doppler wind LiDARs, as detailed in Sect. 4. Indeed, the procedure requires as input accumulation time, LiDAR range gate, and the total number of gates, and generates the optimal angular resolution as output. Similar optimization procedures can be developed for other measurement techniques indeed. The generality of the LiSBOA is now highlighted throughout the manuscript. At L 130, it is reported: "The scope of this work is to define a methodology to post-process scattered data of a turbulent velocity field measured through a scanning Doppler wind LiDAR (or, eventually, other remote sensing instruments) to calculate mean, standard deviation and even higher-order statistical moments on a

Cartesian grid." At L 676, it is reported: "Also, some modifications could extend the application of the LiSBOA to other remote sensing instruments, such as sodars and scanning radars."

*b) Line 22: One primary systematic error of cup anemometers is overspeeding (Busch and Kristensen, 1976), which should be mentioned here.*

R: We thank the Reviewer for pointing this out. At L 54, it is now reported: "Besides their simplicity, mechanical anemometers are affected by errors due to the flow distortion of the supporting structures, drawbacks under harsh weather conditions (Mortensen, 1994), and overspeeding (Busch and Kristensen, 1976)".

c) It would be beneficial to add a list of symbols to the start of the article, as is done for the companion paper. Not being overly-familiar with the Barnes analysis technique, the symbol meaning was very difficult to keep track of.

R: We added the list of symbols as suggested by the Reviewer.

d) Line 46: Fog and low clouds should be added to this list of adverse weather conditions.
R: At L 79, it is now reported: "Besides the mentioned capabilities, LiDARs present some important limitations, such as reduced range in adverse weather conditions (precipitations, heavy rain, fog, low clouds or low aerosol concentration) (Liu et al. (2019), Mann et al. (2018))".

*e)* Line 65: Should be 'horizontal homogeneity of the mean flow', as the turbulent portion may vary over the scanning volume.

R: The Reviewer is right; the manuscript has been revised accordingly (L 97): "In this regard, Eberhard et al. (1989) re-adapted the post-processing of the velocity azimuth display (VAD) scans (Lhermitte, 1969; Wilson, 1970; Kropfli, 1986) to estimate all the components of the Reynolds stress tensor by assuming horizontal homogeneity of the mean flow within the scanning volume, which can be a limiting constraint for measurements in complex terrains (Frisch, 1991; Bingöl, 2009)".

*f) Line* 71-76: *These sentences seem out of place, considering the previous paragraph discussed methods to estimate turbulence. They would better fit in there.*

R: This part of the text has been rephrased as (now L 96): "Besides the mean field, the calculation of higher-order statistics from LiDAR data to investigate atmospheric turbulence is still an open problem. In this regard, Eberhard et al. (1989) re-adapted the post-processing of the velocity azimuth display (VAD) scans (Lhermitte, 1969; Wilson, 1970; Kropfli, 1986) to estimate all the components of the Reynolds stress tensor by assuming horizontal homogeneity of the mean flow within the scanning volume, which can be a limiting constraint for measurements in complex terrains (Frisch, 1991; Bingöl, 2009). Range height indicator (RHI) scans were used to detect second-order statistics (Bonin et al., 2017), spectra, skewness, dissipation rate of the velocity field, and even heat flux (Gal-Chen et al., 1992). Recently, in the context of wind radar technology, but readily applicable to LiDARs as well, a promising method for the estimation of the instantaneous turbulence intensity (i.e. the ratio between standard deviation and mean of streamwise velocity) based on the Taylor hypothesis of frozen turbulence was proposed by Duncan et al. (2019). More advanced techniques exploit additional information of turbulence carried by the spectrum of the back-scattered LiDAR signal (Smalikho, 1995). However, this approach requires the availability of LiDAR raw data, which is not generally granted for commercial LiDARs. For a review on

turbulence statistical analyses through LiDAR measurements, the reader can refer to Sathe and Mann (2013). Another typical scanning strategy to obtain high-frequency LiDAR data consists in performing scans with fixed elevation and azimuthal angles of the laser beam while maximizing the sampling frequency (Mayor et al., 1997; O'Connor et al., 2010; Vakkari et al., 2015; Frehlich and Cornman, 2002; Debnath et al., 2017b; Choukulkar et al., 2017; Lundquist et al., 2017)".

g) Line 123: Throughout this discussion, it is clear that the Barnes analysis can only be applied to a scalar field, and not a vector field such as a wind field composed of a 3-D vector (u, v, and w). The method described here is to be applied to Doppler wind lidar measurements, which is actually the projection of the 3-D wind vector along the beam. Please discuss limitations, assumptions, and other issues that need to be considered when applying this technique, which was developed for scalar quantities, to analyzing any components of a vector wind field. Within this discussion, it must be considered that a Doppler lidar only measures the wind vector reprojected to be along the radial, thus for a scan it is actually sensitive to varying components of the wind based on the azimuth and elevation.

R: We thank the Reviewer for this important comment, which deserves to be clarified. As it is currently formulated, the LiSBOA deals with a scalar field, which can be the LiDAR radial velocity or, for scans with negligible elevation angle, negligible vertical velocity and under the assumption of constant wind direction, the horizontal equivalent velocity (see e.g. Zhan et al. 2019, 2020). In principle, the LiSBOA can be extended to vector fields where the various scalar components are simultaneously analyzed and other constraints can be added to the optimization process, such as mass conservation through a divergence-free velocity field for incompressible flows. For the sake of clarity, at L 568 it is now reported: "The latter states that, for small elevation angles (i.e.  $\beta \ll 1$ ) and under the assumptions of negligible vertical velocity compared to the horizontal component (i.e.  $|w| \ll \sqrt{u^2 + v^2}$ ) and uniform wind direction,  $\theta_w$ , a proxy for the streamwise velocity can be calculated as:

$$u \sim u_{LOS} / [cos(\theta - \theta_w) cos\beta]."$$

At L 673, it is now added: "It is noteworthy that the LiSBOA is currently formulated for a single scalar field, namely a velocity component (radial or equivalent horizontal). However, in principle this procedure can be extended to vector fields, such as a fully 3D velocity fields. Furthermore, other constraints can be added for the optimal scanning design, such as imposing a divergence-free velocity field for incompressible flows."

*h)* Line 354: Based on this figure, it seems like the flow characteristics are critical to design the optimal scan. However, often the flow characteristics are not known and are the objective of the study is to quantify the flow characteristics. In the absence of knowing the flow characteristics, how does one determine the ideal scan and retrieval of statistics especially for flows that have not been studied extensively previously?

R: We understand the importance of this comment. We added a new section (Sect. 6) to discuss the sensitivity of the LiSBOA to the uncertainty of the inputs, and issues connected with the non-stationarity of the wind. See, in particular, L 627-672.

*i)* Line 391: Given the strict limitations of being able to use the technique to reconstruct the flow and its properties, it is a bit of a stretch to state that this is likely to become a standardized method for LiDAR data collection and analysis. While the proposed technique seems highly

valuable to investigate the properties of low-altitude atmospheric flows with LiDAR, the chief limitation of needing to assume an unvarying mean flow (wind direction and speed unchanging) over the course of the time period of analysis will significantly limit when this technique can be used, especially given that conditions within the PBL are often highly variable particularly during transitional time periods (around sunrise and sunset, most notably). This significant limitation will limit any use for routine automated analysis, but it will certainly be useful in select case studies. This significant limitation must be more directly discussed within the manuscript, to mitigate the misuse of this technique for analyzing flows when it is not appropriate to use. By using this during time periods when the mean conditions are even slightly changing, there will be a significant overestimate of any variance in the flow and other properties (length scales, etc) will also not be representative of the true properties of the flow.

R: We agree with the Reviewer and we have rephrased the conclusions accordingly. We now specify that our vision is to apply the LiSBOA not necessarily to datasets collected continuously under roughly stationary wind conditions, rather for data collected even during non-continuous periods yet under similar wind conditions, such as wind speed and direction at a reference height, and atmospheric stability regime. This kind of study can be performed through a cluster analysis of the available datasets, as we have done in previous works of LiDAR measurements of wind turbine wakes (see e.g. Iungo et al. 2018, Zhan et al, 2019 and 2020). In these works, the LiDAR data were clustered based on incoming wind speed at hub height of the turbines and Bulk Richardson number or turbulence intensity. With the cluster analysis, we can achieve better statistical stationarity. Furthermore, in the companion paper (Part II, Sect. 3), we have performed a sensitivity analysis to the degree of non-stationarity of the inflow, showing that the method is robust even for moderately changing wind conditions. At L 189, it is now reported: "It is further assumed that the field f is ergodic in time. In practice, ergodic data can be obtained by selecting samples collected for a temporal window exhibiting stationary boundary conditions or, more generally, through a cluster analysis of discontinuous data (Machefaux et al., 2016a; Bromm et al., 2018; Iungo et al, 2018; Zhan et al., 2019, 2020)." The new Sect. 6 of Part I discusses this issue in detail (L 612-626).

**Editorial Corrections**

- a) Line 26: Should be volume of  $0.01 \text{ m}^3$ .
- R: The error has been fixed (L 59).

*b)* Line 36: This ("In particular . . . anemometry") is an incomplete sentence.
R: The sentence now continues until "technology (Emeis, 2010)" (L 70).

[revised manuscript text omitted]

---

## Author Comment (AC2) · 25 Nov 2020

**Reply to the comments provided by Anonymous Referee #2 on the manuscript amt-2020-227 entitled "LiSBOA: LiDAR Statistical Barnes Objective Analysis for optimal design of LiDAR scans and retrieval of wind statistics. Part I: Theoretical framework" by S. Letizia, L. Zhan and G.V. Iungo**

The authors sincerely thank the referee for the thorough review and the detailed comments. Our replies are reported in the following. References to pages and lines are based on the revised marked-up manuscript.

**Comments:**
*This manuscript presents a new algorithm to retrieve wind speed and turbulence intensity from Doppler lidar measurements. The algorithm is based on Barnes objective analysis, which was developed to interpolate unevenly spaced observations of a scalar field over a regular grid. Here, this method is extended to 3D-fields, i.e. radial velocity from Doppler lidar. Furthermore, the new algorithm is used to optimise Doppler lidar scan strategy for optimal retrieval. On the other hand, the algorithm requires substantial preliminary information about the flow in the area of interest as well as stationary conditions during the period when the scans are conducted, which limits the use of the algorithm. Overall, I find this work within the scope of AMT, but there are several major issues that need to be addressed first.*
R: We appreciate the positive feedback and constructive comments. The manuscript discussion has been extended to clarify the role of the input parameters and statistical stationarity of the dataset for the calculation of the flow statistics and optimal design of LiDAR scans through the LiSBOA. We have now devoted a new section (Sect. 6) to discuss the applicability of the proposed techniques to non-stationary field measurements and test the sensitivity of the LiSBOA to the various inputs. For instance, we suggest the application of the LiSBOA on subsets of the original LiDAR dataset obtained through cluster analysis. The latter can be performed based on relevant flow parameters, such as wind speed, direction, and atmospheric stability regime at a reference height. This approach, which was already applied for the analysis of LiDAR measurements of wind turbine wakes (see e.g. Machefaux et al. 2016, Bromm et al. 2018, Iungo et al. 2018, Zhan et al. 2019, 2020), it can enhance statistical stationarity of the LiDAR datasets.

**Specific comments:**

*1.      Benefits of the new algorithm are not demonstrated. Synthetic data is used to show that this method can retrieve wind speed and turbulence intensity (defined as u'/U), but these parameters can be retrieved with much simpler methods and less previous information about the flow from Doppler lidar data. How much does the new algorithm improve the retrieval?*
R: The LiSBOA is not only a tool for retrieval of wind-flow statistics, rather a tool for the optimal design of LiDAR scans to generate flow statistics with a prescribed accuracy (mean, turbulence intensity, and even higher-order statistics). The methodology to calculate flow statistics is only a sub-part of the proposed statistical framework. Furthermore, by leveraging the Peterson-Middleton criterion, we can identify spatial regions of the measurement domain where the flow statistics are fully-resolved with a prescribed spatial resolution (wavelength). To the best of the authors' knowledge, we are not aware of a such holistic tool for the design and post-processing of LiDAR scans. The LiSBOA is firstly assessed against synthetic data for idealized isotropic turbulence to provide a theoretical function of the error response of the LiSBOA. Subsequently, it has been

tested against LES and real LiDAR data of wind turbine wakes showing excellent agreement with the experimental data, such as SCADA of the wind turbines. At L 142, it is now reported: "It will be shown that the revisited Barnes scheme offers several advantages compared to the above-cited techniques for LiDAR data analysis: *i)* it allows to explicitly select the cut-off wavenumber to filter out small-scale variability while retaining relevant modes in the flow field; *ii)* the distance-based weighting function provides smoother fields than linear interpolation or box-averaging, while still being simpler and computationally inexpensive compared to more sophisticated techniques (e.g. optimal interpolation, variational methods); *iii)* it provides guidance for the optimal design of LiDAR scans to investigate specific wavelengths in the flow. On the other hand, the procedure requires estimates of input parameters for the flow under investigation and the LiDAR system used. In case these parameters cannot be obtained from existing literature or preliminary tests, then a sensitivity study on the variability of the LiSBOA results to the input parameters can be carried out. ". At L 681, it is reported: "This LiDAR Statistical Barnes Objective Analysis (LiSBOA) provides a theoretical framework to quantify the response in the reconstruction of the velocity statistics as a function of the spatial wavelengths of the velocity field under investigation and quantification of the sampling error". At L 686, it is reported: "The LiSBOA framework provides guidelines for the optimal design of scans performed with a scanning Doppler pulsed wind LiDAR and calculation of wind velocity statistics".

*2.      Can this method be used to retrieve information about turbulence in the inertial sub-range, or is it limited to the outer scale of the velocity spectrum?*
R: We thank the Reviewer for this comment. The LiSBOA provides estimates of the statistical errors given an ideal accuracy of the instrument (zero instrumentation error). Any a-priori error should be combined with the sampling error estimated through the LiSBOA (Wheeler and Ganji, 2010). Eventually, the spatial averaging of the LiDAR can be corrected before the application of the LiSBOA, see e.g. a recent paper under review for AMT Puccioni & Iungo 2020. Furthermore, it is now reported (L 410): "It is noteworthy that the estimates of the statistical error obtained through the LiSBOA do not consider other sources of error, such as accuracy of the instruments and spatial averaging due to the LiDAR measuring process (Rye and Hardesty, 1993; O'Connor et al., 2010; Puccioni and Iungo, 2020) Eventually, other error estimates can be coupled with the sampling error estimated through the LiSBOA for a more comprehensive error analysis (Wheeler and Ganji, 2010b).".

*3.      Section 3. It seems that LiSBOA is validated with a velocity field (Fig. 1a), which I cannot imagine being ever observed in the atmosphere. Furthermore, the sampling appears to be completely random points, when a lidar would always observe radial velocity along line of sight, i.e. denser spacing of (radial) observations near the instrument. Is this assessment valid for atmospheric use of LiSBOA? Please consider using the LES data from Part II for the validation. Moving analysis of LES from Part II to Part. I would also shorten Part II, which now has 31 figures.*
R: The synthetic isotropic turbulence data are only used in Sect. 3 to generate the error response of the LiSBOA as a function of the data spacing metric, $\Delta d$, and iteration number in analogy to previous seminal papers on the Barnes scheme theory (Barnes 1994). Subsequently, the LiSBOA is assessed against LES and real LiDAR data of wind turbine wake measurements. For the latter, the results of the LiSBOA are assessed against wind statistics measured at the turbine locations through the SCADA. The section about the LES assessment has been moved to Part I.

*4. L290: "Firstly, it is crucial to estimate the integral quantities of the flow under investigation required for the application of the LiSBOA, such as extension of the spatial domain of interest, characteristic length-scales, integral time-scale, characteristic temporal standard deviation of the velocity, and expected total sampling time, T, which depends on the typical duration of stationary boundary conditions over the domain." How sensitive is LiSBOA to uncertainty in these parameters? How strict is the requirement of stationary boundary conditions? I'm afraid that these requirements will become a severe limitation for the use of LiSBOA.*

R: We thank the Reviewer for this comment, which is now better clarified in the revised manuscript. We now clarify that our vision is to apply the LiSBOA not necessarily to datasets collected continuously under roughly stationary wind conditions, rather for data collected even during non-continuous periods yet under similar wind conditions, such as wind speed and direction at a reference height, and atmospheric stability regime. This kind of study can be performed through a cluster analysis of the available datasets, as we have done in previous works of LiDAR measurements of wind turbine wakes (see e.g. Iungo et al. 2018, Zhan et al, 2019 and 2020). In these works, the LiDAR data were clustered based on incoming wind speed at hub height of the turbines and Bulk Richardson number or turbulence intensity. With the cluster analysis, we can enhance statistical stationarity for the calculation of the flow statistics. Furthermore, in the companion paper (Part II, Sect. 3), we have performed a sensitivity analysis to the degree of non-stationarity of the inflow, showing that the method is robust even for moderately changing wind conditions. At L 189, it is now reported: "It is further assumed that the field $f$ is ergodic in time. In practice, ergodic data can be obtained by selecting samples collected for a temporal window exhibiting stationary boundary conditions or, more generally, through a cluster analysis of discontinuous data (Machefaux et al., 2016a; Bromm et al., 2018; Iungo et al, 2018; Zhan et al., 2019, 2020)." The new Sect. 6 discusses this issue in detail (L 612-626).

*5. L327-329: "The angular resolution of the LiDAR scanning head, can be selected to modify the angular spacing between consecutive line-of-sights (i.e. the data spacing)". Do you propose to use the same spacing for both azimuth and elevation angles? Or use only one PPI scan? Please include both angles in the optimization process.*

R: Thank you for highlighting this point. Different combinations of angular steps for azimuth, $\Delta\theta$, and elevation, $\Delta\beta$, can be tested in the same Pareto front with this procedure. For the LES validation of Sect. 5, the axial symmetry of the wake allows using $\Delta\theta = \Delta\beta$. At L 383, it is also now stated: "The angular resolution of the LiDAR scanning head in azimuth ($\Delta\theta$, for PPIs) or elevation ($\Delta\beta$, for RHIs) or both axis (for volumetric scans), can be selected to modify the angular spacing between consecutive line-of-sights (i.e. the data spacing) and the total sampling period for a single scan, $\tau_s$ (i.e. the number of realizations, $L$)". Equations and figures of Sect. 4 have been revised accordingly.

*6. It seems that instrumental noise is not considered in the algorithm. What is suitable instrumental uncertainty in radial velocity for use in LiSBOA? This is vital information for scan design as sampling time per profile and/or gate length may need to be increased to cover the full area of interest with strong enough signal.*

R: We assume that the data provided to the LiSBOA are quality-controlled, see e.g. Manninen et al. (2016); Beck and Kühn (2017); Vakkari et al. (2019). Indeed, you can see at L 383 and in Fig. 7 that the LiDAR range gate, number of gates, and accumulation time should be provided as inputs

of the LiSBOA. We understand that these inputs might vary due to the site/atmospheric conditions. However, different scenarios can be easily simulated with the LiSBOA by using different values for the various inputs. At L 432, it is now reported: "The interested reader is referred to Manninen et al. (2016); Beck and Kühn (2017); Vakkari et al. (2019) for more information on data quality check.". Furthermore, at L 666 it is now reported: "To conclude, we acknowledge that the technical specifications required by LiSBOA (namely $\tau_a, N_r, \Delta r$) are dependent on the specific LiDAR system used, the contingent atmospheric conditions and the best practices followed by the user. Since these parameters are greatly case-dependent, they will not be discussed further in this context. In general, the selection of the accumulation time and gate length is a trade-off between the need to achieve a target maximum range, while keeping a sufficiently fine radial resolution and sufficient intensity of the back-scattered LiDAR signal. In case of uncertain environmental conditions, it is recommended to check before the deployment the influence of selected combinations of $\tau_a, N_r, \Delta r$ on the Pareto front.".

7.      *Fig. 7. Thank you for the diagram. Please also provide code for the optimization process using these inputs.*
R: The code has now been made available.

8.      *L395 The code is not available where stated. Please include the code as supplementary information so that a copy is permanently archived.*
R: The code has now been made available.

[revised manuscript text omitted]
. ~~As described in ?, the optimization problem of a LiDAR scan consists of minimizing two cost functions. The first cost function, $\epsilon^I$, represents the percentage of grid nodes for which the Petersen-Middleton constraint applied to the smallest half-wavelength of interest, $\Delta n_0$, is not satisfied. In other words, it represents the percentage of the measurement domain that is not sampled with adequate data spacing so that aliasing of the LiDAR data may occur. The second cost function, $\epsilon^{II}$, is the standard deviation of the sample mean and quantifies the temporal statistical uncertainty due to the finite number of scan repetitions, $L$, allowed within the total sampling time, $T$.~~

[revised manuscript text omitted]

---

## Author Response (AR3)

**Reply to the comments provided by the Anonymous Referee #2 on the manuscript amt-2020-227 entitled "LiSBOA: LiDAR Statistical Barnes Objective Analysis for optimal design of LiDAR scans and retrieval of wind statistics. Part I: Theoretical framework", by S. Letizia, L. Zhan and G.V. Iungo**

The authors thank the Reviewer for the constructive comments. Our replies are reported in the following. References to pages and lines are based on the latest marked-up manuscript.

**Comments:**

*I would like to thank the authors for their reply to my comments, although some of the comments were not fully addressed.*
R: We thank the Referee for thoroughly reviewing our rebuttal. We have now implemented the suggested modifications.

**Specific Comments:**

*1.       On comment 1, I agree that this method has potential to improve the current techniques for lidar data analysis. However, this has not been shown. It should be quite straightforward to use the LES data set to show that the LiSBOA retrieved winds and turbulence intensity are closer to the true wind field than other methods.*
R: We agree with the Reviewer and added a paragraph in Sect. 6 (L 659) and Fig. 20 where mean streamwise velocity and streamwise turbulence are retrieved with LiSBOA, Delaunay triangulation, linear interpolation, and window average. The analysis shows that LiSBOA enables the largest spatial coverage for the retrieved flow statistics for a given LiDAR scan (double than the window average technique). Overall, all the methods have similar accuracy in terms of MAPE for the retrieval of the mean velocity, yet LiSBOA shows the highest accuracy. Furthermore, LiSBOA does not show artifacts in the retrieval of the turbulence intensity, such as unexpected peaks or enhanced turbulence intensity, as for the remaining methods.

*2.       The response to comment 2 does not respond to the question. If the outer scale non-turbulent eddies (see e.g. Fig. 1 in O'Connor et al., 2010) cannot be filtered out with this method, it should be stated in the manuscript.*
R: That's correct. At L 403, it is now reported: "Furthermore, LiSBOA allows calculating velocity statistics including contributions of eddies with different sizes, which span from the largest eddy advected within the total sampling time, to the smallest eddy detectable for a given accumulation time (Puccioni & Iungo 2020). Therefore, a careful pre-processing of the LiDAR data should be eventually performed to remove contributions due to non-turbulent mesoscale eddies (Högström et al.; 2002, Metzeger et al., 2007; O'Connor et al., 2010)."

*3.      On comment 5, the authors clarify the use of equal spacing for azimuth and elevation angles in the optimisation in this case. However, my request to include both angles in the optimisation process is not implemented. Furthermore, I note that in Part II the authors use a scan schedule where they first set elevation angle spacing to [5, 6, 7, 8, 10, 12, 15] and then optimise the azimuth angle. In Part II also the sampling time per profile is decided prior to applying the LiSBOA optimisation. So it seems that only a subset of the scan parameters are provided through LiSBOA, though optimal scan design is one of the main arguments for using LiSBOA.*

R: LiSBOA can optimize all the parameters involved with the scanning strategy. Further constraints can be added based on physics knowledge and existing literature. The choice of equal azimuth and elevation resolution was based on the physical consideration that the wavelength in $y$ and $z$ are comparable. However, we agree that exploring combinations of the ratio $\Delta\beta/\Delta\theta$ other than 1 can be interesting. We added two additional Pareto fronts with $\frac{\Delta\beta}{\Delta\theta} = 0.5$ and $\frac{\Delta\beta}{\Delta\theta} = 2$ in Fig. 11. At L 540 it is now reported: "For the optimization of the LiDAR scan, the LiDAR angular resolution, $\Delta\theta$, is evenly varied, for a total number of 7 cases, from 0.75° to 4°, whereas three values of the ratio $\Delta\beta/\Delta\theta$, namely 0.5, 1 and 2, are tested separately". Also, at L 544 it is now reported: "Changing the ratio $\Delta\beta/\Delta\theta$ affects the optimal $\Delta\theta$ (circled in black in Fig. 11); however, it has a negligible effect on the magnitude of the optimal $\epsilon^I$ and $\epsilon^{II}$. For the rest of the discussion, we focus on the setup $\Delta\beta/\Delta\theta = 1$, as suggested by Fuertes Carbajo and Porté-Agel (2018)." Part II has also been revised according to the suggestions of the Reviewer to clarify the choice of the elevation angle and the role of LiSBOA in the scan design process.

*4.      On comment 6, the reply to comment 2 seems more relevant, i.e. zero instrument error appears to be assumed. This is a major drawback, as lidar measurement will always have some instrumental error. However, by filtering out small-scale variability LiSBOA possibly allows using relatively noisy measurements, increasing the amount of useful data. Therefore, the effect of instrumental error must be investigated in more detail. Please use the LES data to determine maximum acceptable instrumental uncertainty in radial wind measurement. It should be relatively simple to add random noise to the lidar simulator (Eq. 17 in the revised manuscript) to find limits for acceptable noise. This will also determine minimum CNR threshold, which is essential for determination of sampling time per profile.*

R: We agree with the Reviewer that the effects of noise of LiDAR data for the retrieval of the statistics is an interesting topic to be investigated; however, this topic is out of the scope for this manuscript. Although it is true that the spatial averaging connected with the Barnes scheme may be beneficial for the suppression of short-wavelengths noise from the mean field (Barnes 1973, Pauley and Wu, 1991, Barnes 1994), effects on higher-order statistics are more unpredictable because they are affected by the noise variance, pdf, and its modeling as a function of the SNR of the LiDAR signal, the specific LiDAR system used, probe length, and variability on the background turbulence ( see e.g. Frehlich and Kavaya, 1991). This is clearly the scope for future research.

*5.      On comments 7 and 8: The code is still not available. L682-683: "Code availability. The LiSBOA algorithm is implemented in a publicly available code which can be downloaded at the following URL: https://www.utdallas.edu/windflux/." This seems to be a group home page. Underneath it there is a page "Software/Datasets": https://www.utdallas.edu/windflux/software-datasets/ but even that does not contain the codes used here. Please use a permanent repository for*

*the code or include it as supplement so that it is permanently stored. Group home page is ok for keeping the latest version available, but it is not a permanent storage.*

R: We uploaded the codes for scan design and statistics reconstruction on GitHub [https://github.com/UTD-WindFluX/LiSBOA](https://github.com/UTD-WindFluX/LiSBOA). We made sure that the URL is permanently available now at https://www.utdallas.edu/windflux/software-datasets/. We apologize for the inconvenience with the previous upload.

**References**

Barnes, S.L.: Application of Barnes Objective Analysis scheme. Part III: tuning for minimum error, *J. Atmos. Ocean. Tech*., 11, 1449–1458, 1994.

Barnes, S.L.: Mesoscale Objective Map Analysis Using Weighted Time-Series Observations, *NOAA Technical Memorandum ERL NSSL-76062*, National Severe Storms Laboratory Norman, OK, 1973.

Frehlich, R., Kavaya, M.: Coherent laser radar performance for general atmospheric refractive turbulence, *Applied Optics*, 30 (36), 5325-5352, 1991.

Carbajo Fuertes, F. and Porté-Agel, F.: Using a Virtual Lidar Approach to Assess the Accuracy of the Volumetric Reconstruction of a Wind Turbine Wake, *Remote Sens.-Basel*, 10, 2018.

Högström, U., Hunt, J. C. R., and Smedman, A. S.: Theory and measurements for turbulence spectra and variances in the atmospheric neutral surface layer, *Bound.-Lay. Meteor.,* 103, 101-124, 2002.

Metzger, M., McKean, B.J., Holmes, H.: The near-neutral atmospheric surface layer: turbulence and non-stationarity, *Phil. Trans. R. Spc. A*, 365, 859-876, 2017.

O'Connor, E. J., Illingworth, A. J., Brooks, I. M., Westbrook, C. D., Hogan, R. J., Davies, F., and Brooks, B. J.: A Method for Estimating the Turbulent Kinetic Energy Dissipation Rate from a Vertically Pointing Doppler Lidar and Independent Evaluation from Balloon-Borne In Situ Measurements, *J. Atmos. Ocean. Tech.,* 27, 1652-1664, 2010.

Pauley, P. M. and Wu, X.: The Theoretical, Discrete and Actual Response of the Barnes Objective Analysis Scheme for One- and Two-Dimensional Fields, *Mon. Weather Rev*., 118, 1145–1164, 1990.

Puccioni, M., Iungo, G.V. Spectral correction of turbulent energy damping on wind LiDAR measurements due to range-gate averaging. *Atmos. Meas. Tech. Discussions (accepted for publication*, 1-28, 2020.